# Hamiltonian Score Matching and Generative Flows

**Peter Holderrieth**
MIT CSAIL
phold@mit.edu

**Yilun Xu**
NVIDIA
yilunx@nvidia.com

**Tommi Jaakkola**
MIT CSAIL
tommi@csail.mit.edu

## Abstract

Classical Hamiltonian mechanics has been widely used in machine learning in the form of Hamiltonian Monte Carlo for applications with *predetermined* force fields. In this work, we explore the potential of deliberately designing force fields for Hamiltonian ODEs, introducing Hamiltonian velocity predictors (HVPs) as a tool for score matching and generative models. We present two innovations constructed with HVPs: *Hamiltonian Score Matching (HSM)*, which estimates score functions by augmenting data via Hamiltonian trajectories, and *Hamiltonian Generative Flows (HGFs)*, a novel generative model that encompasses diffusion models and flow matching as HGFs with zero force fields. We showcase the extended design space of force fields by introducing Oscillation HGFs, a generative model inspired by harmonic oscillators. Our experiments validate our theoretical insights about HSM as a novel score matching metric and demonstrate that HGFs rival leading generative modeling techniques.

## 1 Introduction

Hamiltonian mechanics is a cornerstone of classical physics, providing a powerful framework for analyzing the dynamics of physical systems [33, 4]. The Hamiltonian formalism has been widely applied in machine learning and Bayesian statistics via Hamiltonian Monte Carlo (HMC) [14, 36, 8]. In this setting, the goal is to sample from a probability distribution $\pi$ whose density $\pi(x)$ is known up to a normalization factor. In HMC, one interprets $\nabla \log \pi(x)$ as a force function and plugs it into a Hamiltonian ODE to construct a fast-mixing Markov chain exploring the data space quickly. This makes HMC one of the most powerful sampling algorithms to date [36, 23].

In generative modeling, the density $\pi(x)$ is unknown, only data samples $x_1, \ldots, x_n \sim \pi$ are given, and the goal is to learn to generate novel samples from $\pi$. Current state-of-the-art models are based on diffusion [41, 42, 45, 21] and enjoy widespread success in image generation [39], molecular generation [11], and robotics [10]. Diffusion models *learn* the score function $\nabla \log \pi_\sigma(x)$ for a range of noise scales $\sigma$ via denoising score matching (DSM) [48]. This enables one to subsequently generate high-quality samples by following a stochastic differential equation [45].

In light of the success of HMC, it is natural to ask whether the Hamiltonian formalism can also improve generative models or provide novel insights into their construction. Previous works have exploited the connection between Hamiltonian physics and generative modeling for specific force fields [13]. However, these works usually consider particular (fixed) force fields and stay within the diffusion framework.

More recently, flow-based generative models such as flow matching have enabled scalable training of continuous normalizing flows (CNFs) [31, 32]. These ODE-based models allow to craft first-order ODEs transforming arbitrary distributions from one to another. Inspired by these successes, we consider the Hamiltonian ODE as a Neural ODE [7]. We show that we can marginalize out velocity

38th Conference on Neural Information Processing Systems (NeurIPS 2024).

distributions and then follow backward ODEs that faithfully recover data distributions. Importantly, we provide an associated theorem that holds for any force field. With the growing success of generative models in physical sciences [11, 49, 1], it is striking that most approaches do not use existing known force fields - sometimes leading to physically implausible results [1]. A framework that allows to reason natively about force fields in the context of generative models would be very promising for such applications [2, 12]. This work aims to build towards such a deeper integration.

We explore the intricate relationship between Hamiltonian dynamics, force fields, and generative models. Specifically, we make the following contributions:

1. **Hamiltonian velocity predictor:** We introduce the concept of a Hamiltonian Velocity Predictor (HVP) and show the utility of HVPs for score matching and generative modeling (Section 3).

2. **Hamiltonian score discrepancy (HSD):** We introduce and validate Hamiltonian score discrepancy (HSD), a novel score matching metric based on HVPs and a corresponding score matching method (Section 4).

3. **Hamiltonian generative flows (HGFs):** We show that the location marginal of a Hamiltonian ODE is generated via the Hamiltonian Velocity Predictor (Section 5). This leads to a novel generative model generalizing diffusion models and flow matching (Section 6).

4. **Oscillation HGFs:** As special HGFs, we study Oscillation HGFs, a simple generative model rivaling the performance of diffusion models due to in-built scale-invariance (Section 7).

## 2 Background

### 2.1 Hamiltonian Dynamics

The inspiration of using Hamiltonian dynamics in machine learning comes from considering a data point $x \in \mathbb{R}^d$ as coordinates of an object in $\mathbb{R}^d$. Such an object also has a velocity $v \in \mathbb{R}^d$. Let $\pi$ be a probability distribution with density function $\pi : \mathbb{R}^d \to \mathbb{R}_{\geq 0}$. Assuming unit mass, the energy of such an object is given by its Hamiltonian [33, 4], defined by:

$$H(x,v) = U(x) + \frac{1}{2}\|v\|^2, \tag{1}$$

where $U : \mathbb{R}^d \to \mathbb{R}$ is a potential function. The key idea behind using Hamiltonian dynamics in the context of probabilistic modeling and sampling is to set the potential function as negative log-likelihood, i.e., $U(x) = -\log \pi(x)$. One defines the *Boltzmann-Gibbs distribution* $\pi_{BG}$ then as:

$$\pi_{BG} = \pi \otimes \mathcal{N}(0, \mathbf{I}_d), \quad \pi_{BG}(x,v) = \exp(-H(x,v))/Z = \pi(x)\mathcal{N}(v; 0, \mathbf{I}_d),$$

i.e. the product distribution of the data distribution and normal distribution. In particular, one can easily draw a sample $z$ from $z \sim \pi_{BG}$ by sampling $x \sim \pi, v \sim \mathcal{N}(0, \mathbf{I}_d)$ and setting $z = (x,v)$.

Hamiltonian dynamics describe how an object described by $z = (x_0, v_0)$ evolves over time. It is defined by the ODE [4]

$$(\frac{d}{dt}x(t), \frac{d}{dt}v(t)) = (v(t), -\nabla U(x(t))) = (v(t), \nabla \log \pi(x(t))), \tag{2}$$

$$(x(0), v(0)) = z \tag{3}$$

i.e. the change of location is the velocity and the change of velocity is the force (here, equals acceleration as we assume unit mass). Let $\varphi : \mathbb{R}^{2d} \times \mathbb{R} \to \mathbb{R}^{2d}, (z, t) \mapsto \varphi_t(z)$ be the corresponding flow, i.e. the function $t \mapsto \varphi_t(z)$ is a solution to the above ODE with starting point $z$.

As one would expect from physics, Hamiltonian dynamics $\varphi_t$ preserve the energy of a system, i.e. $H(\varphi_t(x,v)) = H(x,v)$ for all $t, x, v$ (see proof in Appendix B.1). This physical intuition translates into the fact that Hamiltonian dynamics preserve the Boltzmann-Gibbs distribution, i.e.

$$Z \sim \pi_{BG} \Rightarrow \varphi_t(Z) \sim \pi_{BG} \text{ for all } t \geq 0 \tag{4}$$

We include a derivation of this well-known statement in Appendix C.1 as it is so central to this work.

## 2.2 Score Matching

The goal of score matching is to learn the *score function* $\nabla \log \pi$ from data samples $x_1, \ldots, x_n \sim \pi$. As the score function naturally appears in the Hamiltonian ODE (see Equation (2)), we interpret it as a force function and denote a parameterized score model by $F_\theta : \mathbb{R}^d \to \mathbb{R}^d$. A natural approach to fitting $F_\theta$ is to minimize the mean squared error between $F_\theta$ and the true score weighted by their likelihood under $\pi$. This leads to the *explicit score matching loss* [25] given by

$$L_{\text{esm}}(\theta; \pi) = \mathbb{E}_{x \sim \pi} \left[ \frac{1}{2} \|\nabla \log \pi(x) - F_\theta(x)\|^2 \right]. \tag{5}$$

This loss cannot be minimized directly as one does not have access to $\nabla \log \pi$ and various score-matching methods differ in how they circumvent not having access to $\nabla \log \pi$ (see Appendix A for a detailed overview).

A different approach to score matching is to slightly modify the objective by adding Gaussian noise to the data distribution $\pi$ to get the noisy distribution $\pi_\sigma(x) = \int \mathcal{N}(x; x_0; \sigma^2 \mathbf{I}_d) \pi(x_0) dx_0$ [48]. The objective can then be expressed as *denoising score matching*:

$$L_{\text{dsm}}(\theta; \pi_\sigma) = \mathbb{E}_{x \sim \pi, \epsilon \sim \mathcal{N}(0, \mathbf{I}_d)} \left[ \|F_\theta(x + \sigma\epsilon) + \frac{\epsilon}{\sigma}\|^2 \right] = L_{\text{esm}}(\theta; \pi_\sigma) + C_\sigma \tag{6}$$

for a constant $C_\sigma$. Noised data distributions naturally appear in diffusion models, and the denoising score-matching objective, therefore, became the state-of-the-art method to train diffusion models. However, denoising score matching suffers from high variance leading to long training times as well as computationally expensive sampling [44, 52]. In addition, it would be an unreasonable choice for an application where it is important to learn the original data distribution.

## 3 Hamiltonian Velocity Predictors

The Hamiltonian ODE has been widely used in Bayesian statistics in the form Hamiltonian Monte Carlo [14, 36]. However, in such settings, it is assumed that one has access to the score $\nabla \log \pi$ (=force), and the goal is to sample from $\pi$ (by sampling from $\pi_{BG}$). In machine learning tasks, the inverse is true. For such tasks, one has access to data samples $x_1, \ldots, x_n \sim \pi$ and the goal is to (1) learn $\nabla \log \pi$ (score matching) or (2) create new samples $x \sim \pi$ (generative modeling) - or both.

**Parameterized Hamiltonian ODEs (PH-ODEs).** This inspires the definition of a *parameterized Hamiltonian ODE* (PH-ODE). A PH-ODEs consists of two components:

1. **Initial distribution $\Pi$:** The starting condition $z = (x, v) \sim \Pi$ is distributed according to a joint *location-velocity distribution* $\Pi$ such that its marginal over $x$ is $\pi$:

$$\int \Pi(x, v) dv = \pi(x) \tag{7}$$

In the default cause, $\Pi$ equals the Boltzmann-Gibbs distribution $\pi_{BG}$, i.e. $\Pi = \pi \otimes \mathcal{N}(0, \mathbf{I}_d)$.

2. **Force field:** The evolution is governed by a parameterized force field $F_\theta : \mathbb{R}^d \times \mathbb{R} \to \mathbb{R}^d$ via:

$$(\frac{d}{dt}x(t), \frac{d}{dt}v(t)) = (v(t), F_\theta(x(t), t)) \tag{8}$$

$$(x(0), v(0)) = (x_0, v_0) = z \tag{9}$$

As we consider $\Pi$ as part of the definition of a PH-ODE, we write with a slight abuse of notation

$$\varphi_t^\theta(z) = (x_t^\theta(z), v_t^\theta(z)) = (x_t^\theta, v_t^\theta) \tag{10}$$

for the solution of the ODE assuming $z = (x_0, v_0) \sim \Pi$. We note that while $F_\theta$ might refer to a $\theta$-parameterized neural network, we can also set it to a fixed known vector field. Both cases will be important.

**Velocity prediction is all you need.** The crucial idea of this work is that one can use PH-ODEs for both score matching and generative modeling by **predicting velocities**. For this, we use an auxiliary family of functions $V_\phi : \mathbb{R}^d \times \mathbb{R} \to \mathbb{R}^d$, here usually a neural network. Let us consider the following velocity prediction loss:

$$L_V(\phi|\theta, t) = \mathbb{E}_{(x,v)\sim\Pi}[\|V_\phi(x_t^\theta, t) - v_t^\theta\|^2] \tag{11}$$

By minimizing the above loss over $\phi$, we train $V_\phi$ to predict the velocity given the location after running the Hamiltonian ODE with starting conditions defined by $\Pi$. For a sufficiently rich class of functions $V_\phi$, the minimizer $V_{\phi^*}$ of the above loss is the expected velocity conditioned on the location (see Appendix D):

$$V_{\phi^*}(x, t) = \mathbb{E}[v_t^\theta | x_t^\theta = x] \tag{12}$$

As we will see, this quantity is "all you need" for generative modeling and score matching. We call $V_{\phi^*}$ the **Hamiltonian velocity predictor (HVP)**. We note that many of flow-based works also implicitly learn to predict velocities [31], although they do not consider them as separate states.

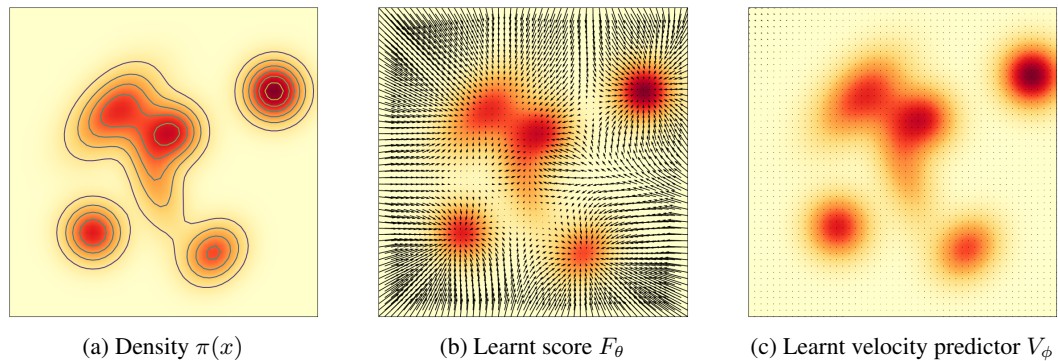

(a) Density $\pi(x)$        (b) Learnt score $F_\theta$        (c) Learnt velocity predictor $V_\phi$

Figure 1: Results of training HSM on a Gaussian mixture. The score vector field faithfully recovers gradients of the density. The optimal velocity predictor is zero everywhere.

# 4 Hamiltonian Score Matching

In this section, we provide a novel score-matching method using PH-ODEs and velocity predictors. We will first connect the score function to a preservation property of Hamiltonian systems (Section 4.1), and then introduce a new score-matching objective derived from this property (Sections 4.2 and Section 4.3).

## 4.1 Characterizing Scores with Hamiltonian Dynamics

The conservation of the Boltzmann-Gibbs distribution $\pi_{BG}$ (see Equation (4)) is the crucial property that enables Hamiltonian Monte Carlo as it allows for proposals of distant states enabling fast mixing of a Markov chain. The inspiration of this work was to take the inverse perspective: rather than considering the preservation of $\pi_{BG}$ a *useful* property of the Hamiltonian ODE, we ask whether it is the *defining* property of the score function. In other words, **is any vector field that preserves the Boltzmann-Gibbs distribution under a PH-ODE automatically the score?** And if yes, **could we train for this property to learn a score?** As we will show, we can characterize the score with an even easier-to-train preservation property solely depending on the velocity predictor.

**Theorem 1.** *Let $T > 0$ and $F_\theta(x)$ a force field. Let $\Pi = \pi_{BG} = \pi \otimes \mathcal{N}(0, \mathbf{I}_d)$. The following statements are equivalent:*

1. ***Score vector field:*** *The force field $F_\theta$ equals the score, i.e. $F_\theta(x) = \nabla_x \log \pi(x)$ for $\pi$-almost every $x \in \mathbb{R}^d$.*

2. ***Preservation of Boltzmann-Gibbs:*** *The PH-ODE with $F_\theta$ preserves the Boltzmann-Gibbs distribution $\pi_{BG}$.*

3. ***Conditional velocity is zero:*** *The velocity given the location after running the PH-ODE with $F_\theta$ is zero if starting conditions $z = (x_0, v_0)$ are sampled from $\pi_{BG}$:*

$$z \sim \pi_{BG} \quad \Rightarrow \quad \mathbb{E}[v_t^\theta(z)|x_t^\theta(z)] = 0 \quad \text{for all } 0 \le t < T \tag{13}$$

A proof can be found in Appendix C and is based on the fact that test functions linear in $v$ have vanishing expectation if Equation (13) holds. We note that the equivalence of conditions (1) and (2) is well-known in statistical physics.

## 4.2 Hamiltonian Score Discrepancy

Condition (3) in Theorem 1 naturally motivates a new way of training $F_\theta$ to approximate a score. Specifically, our goal is **train the force field $F_\theta$ such that its optimal velocity predictor is zero.** By Theorem 1, it necessarily holds $F_\theta = \nabla \log \pi$ in this case. Unfortunately, such a bilevel optimization is not tractable with stochastic gradient descent in general, as it contains two different objectives.

However, a simple trick allows us to convert the above into a single objective. For this, we define the *Hamiltonian Score Matching* loss:

$$L_{\mathrm{hsm}}(\phi|\theta, t) = \mathbb{E}_{z \sim \pi_{BG}}\left[\|V_\phi(x_t^\theta, t)\|^2 - 2V_\phi(x_t^\theta, t)^T v_t^\theta\right] = L_{\mathrm{V}}(\phi|\theta, t) - C(\theta, t) \tag{14}$$

where $C(\theta, t) = \mathbb{E}[\|v_t^\theta\|^2]$. As the value of $C(\theta, t)$ is a constant in $\phi$, it holds that **the optimal velocity predictor is also the unique minimizer of the Hamiltonian Score Matching loss $L_{\mathrm{hsm}}(\phi|\theta, t)$.** However, while the argmin is the same, the actual obtained minimum value is drastically different as the next proposition shows.

**Proposition 1.** *For a sufficiently rich class of functions $(V_\phi)_{\phi \in I}$, it holds that*

$$\mathbb{D}_{hsm}(\theta|t, \pi) := -\min_{\phi \in I} L_{hsm}(\phi|\theta, t) = \mathbb{E}_{z \sim \pi_{BG}}[\|\mathbb{E}[v_t^\theta|x_t^\theta]\|^2] \tag{15}$$

The proof relies on plugging the identity of $V_{\phi^*}$ (see Equation (12)) into Equation (14)) and can be found in Appendix E. By condition (3) in Theorem 1 (see Equation (13)), we want to minimize $D_{\mathrm{hsm}}(\theta|t, \pi)$ in order to learn scores. For this, let's define a distribution $\lambda$ with full support over $[0, T)$ for $T \in \mathbb{R}_{>0} \cup \{\infty\}$). With this, we define the **Hamiltonian score discrepancy (HSD)** as

$$\mathbb{D}_{\mathrm{hsm}}(\theta|\pi) = \mathbb{E}_{t \sim \lambda}\left[\mathbb{D}_{\mathrm{hsm}}(\theta|t, \pi)\right] \tag{16}$$

Note that the discrepancy is defined for an arbitrary (regular) vector field $F_\theta$ - not restricted to scores of probability distributions. By Theorem 1, the discrepancy fulfills all properties that we would expect from a discrepancy to hold: $D(\theta|\pi) \geq 0$ for all $\theta$ and $D(\theta|\pi) = 0$ if and only if $F_\theta = \nabla \log \pi$. We summarize the findings in the below theorem.

**Theorem 2.** *Minimization of the Hamiltonian score discrepancy results in learning the score $\nabla \log \pi$:*

$$\theta^* = \arg\min_\theta \mathbb{D}_{hsm}(\theta|\pi) \Rightarrow s_{\theta^*} = \nabla \log \pi \tag{17}$$

The full proof is stated in Appendix F. The Hamiltonian score discrepancy gives a natural measure of how far a vector field is from the desired score vector field. However, at first, it seems rather abstract. The following proposition shows that minimizing this measure has a very intuitive interpretation. In fact, it is closely connected to the explicit score matching loss $L_{\mathrm{esm}}$ (see Equation (5)).

**Proposition 2** (Taylor approximation of HSM loss)**.** *There exists an error term $\epsilon(t)$ such that*

$$\mathbb{D}_{hsm}(\theta|t, \pi) = 2t^2 L_{esm}(\theta; \pi) + \epsilon(t) \tag{18}$$

*and $\lim_{t \to 0} \frac{1}{t^2}|\epsilon(t)| \to 0$.*

A proof can be found in Appendix G. Intuitively, minimizing the Hamiltonian score discrepancy, therefore, consists of pushing the parabola in Equation (18) down onto the x-axis. The above theorem also indicates the optimal choice of $T$: one should choose $T$ high enough to have a loss value high enough to give signal but low enough to avoid errors due to numerical integration of the ODE.

## 4.3 Hamiltonian Score Matching

Beyond its theoretical value, we can explicitly minimize the HSD, a method we coin *Hamiltonian Score Matching* (HSM). To minimize the HSD, two networks $V_\phi$ and $F_\theta$ can jointly optimize Equation (14). There are two difficulties coming along with this: (1) One has to simulate trajectories. This can be done via Neural ODEs [7] with constant memory. (2) One has to run a min-max optimization. Here, a big toolbox developed for GANs for training stabilization can be used [35, 19]. On the other hand, we hypothesize that HSM has two advantages: (1) every trajectory of HSM gives several points of supervision effectively augmenting our data and (2) we can learn the original ("unnoised") data distribution $\pi$. However, please note that we do *not* propose Hamiltonian Score Matching as a replacement for denoising score matching in diffusion models. Rather, it is a scalable alternative to score matching methods that learn the original ("unnoised") data distribution $\pi$.

## 5 Hamiltonian Generative Flows

Next, we show that training a general velocity predictor of a Hamiltonian ODE is useful even if $F_\theta \neq \nabla \log \pi$. This leads to a generative model that we coin **Hamiltonian Generative Flows** (**HGFs**). As $F_\theta = F$ is fixed and not trained here, we write $(x_t, v_t) = (x_t^\theta, v_t^\theta)$ for the solution of the PH-ODE. Let us denote $\Pi(x, v, t)$ as the distribution of $(x_t, v_t)$ at time $t$ and the **location marginal**

$$\int \Pi(x, v, t) dv = \pi(x, t) \tag{19}$$

The location marginal describes a probability flow starting from our data distribution $\pi = \pi(\cdot, 0)$. It turns out that the optimal velocity predictor is exactly the vector field that generates $\pi(x, t)$.

**Proposition 3.** *Let $\pi_T$ be the distribution such that $x_T^\theta \sim \pi_T$. Let $V_\phi^*$ be the Hamiltonian Velocity Predictor (see Equation* (12)*). Then by sampling $x_T \sim \pi_T$ and running the **velocity predictor ODE***

$$\frac{d}{dt} x(t) = V_{\phi^*}(x, t) \Rightarrow x(0) \sim \pi \tag{20}$$

*backwards in time, we will have $x(0) \sim \pi$, i.e. we can sample from the data distribution $\pi$. More specifically, the optimal velocity predictor $V_{\phi^*}$ generates the probability path $\pi(\cdot, t)$.*

The proof uses the fact that the vector field $G(x, v) = (v, F_\theta(x, t))$ is divergence-free to show that $V_{\phi^*}$ fulfils the deterministic Fokker-Planck equation (see Appendix H). The above proposition allows us to build a generative model by training an HVP. We coin this model *Hamiltonian Generative Flows (HGFs)*. To make this framework tractable, we need two criteria to be fulfilled:

1. **(C1) Forward ODE efficiently computed:** For efficient training, we need to be able to compute $x_t^\theta, v_t^\theta$ efficiently - either via an analytical expression or ODE solvers.

2. **(C2) Initial distribution should be approximately known:** In order to be able to sample the initial point of the ODE faithfully, we need to (approximately) know $\pi(x, T)$.

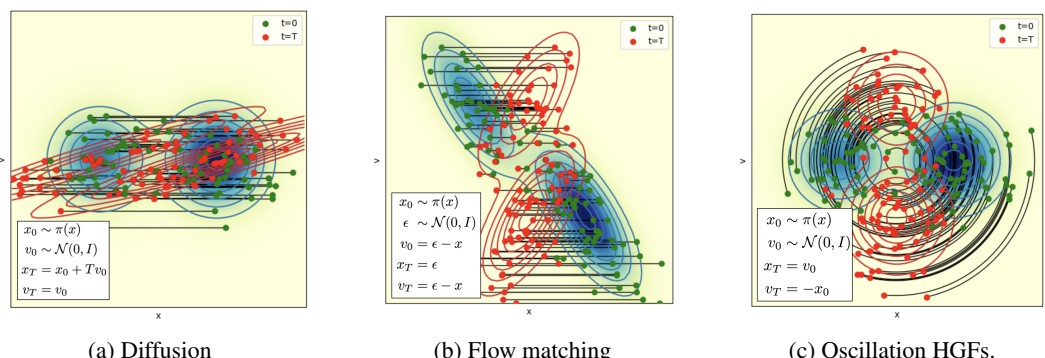

| (a) Diffusion | (b) Flow matching | (c) Oscillation HGFs. |

Figure 2: Evolution of various HGFs in joint coordinate-velocity space from $t = 0$ (blue) to $t = T$ (red) with trajectories (black). Data distribution $\pi(x) = 0.4 * \mathcal{N}(-2, 1) + 0.6 * \mathcal{N}(2, 1)$. Diffusion models and flow matching have zero force fields, i.e. the velocity does not change. Diffusion models do not converge in finite time (here, $T = 3$). The coupled distribution in FM allow for a convergence for $T = 1$. Both distort the joint distribution. Oscillation HGFs only rotate the distribution.

## 6 Diffusion Models and Flow Matching as HGFs with zero force field

### 6.1 Diffusion Models as HGFs

We can recover diffusion models with a variance-preserving (VP-SDE) noising process [45] as a special case of HGFs. If we simply set $\Pi = \pi_{BG}$ and $F_\theta(x) = 0$ - no force applied. In this case, we get that $x_t = x + tv$ and $v_t = v$ leading to the training objective:

$$\mathbb{E}_{x \sim \pi, v \sim \mathcal{N}(0, \mathbf{I}_d)}[\|V_\phi(x_t, t) - v_t\|^2] = \mathbb{E}_{x \sim \pi, \epsilon \sim \mathcal{N}(0, \mathbf{I}_d)}[\|V_\phi(x + t\epsilon, t) - \epsilon\|^2]$$

which equals denoising score matching (Equation (6)) with score model $-tV_\phi(x, t) = \nabla \log \pi_{\sigma(t)}$ with $\sigma(t) = t$. In this case, Hamiltonian Generative Flows correspond to training a diffusion model

and the velocity predictor corresponds to a *denoising network* (often denoted as $\epsilon_\theta$ in DDPMs [21]). It then holds $x_T \sim \pi_T \approx \mathcal{N}(0, \sigma^2(t)\mathbf{I}_d)$ and the velocity predictor ODE then reduces to the well-known probability flow ODE formulation of diffusion models with noise schedule $\sigma(t) = t$:

$$x_T \sim \pi_T \approx \mathcal{N}(0, \sigma^2(t)\mathbf{I}_d), \quad \frac{d}{dt}x(t) = -\dot{\sigma}(t)\sigma(t)\nabla_x \log \pi_{\sigma(t)}(x) = V(x(t), t)$$

In fact, the above is a universal way of modeling diffusion models [26]. In other words, **diffusion models are a special case of HGFs for the zero-force field**. In this perspective, different diffusion models correspond to different time rescaling and preconditioning of the network. The location marginals $\pi(x, t)$ only fully converge to a Gaussian in the limit of $t \to \infty$ (see Figure 2).

## 6.2 Flow Matching as HGFs

Flow matching with the CondOT probability path [31], a current state-of-the-art generative model, can be easily considered an HGF model. As in diffusion, let us consider the zero force field $F_\theta$ and let's consider a coupled initial distribution $\Pi$:

$$x \sim \pi, \quad v = \epsilon - x, \quad \epsilon \sim \mathcal{N}(0, \mathbf{I}_d) \tag{21}$$

Similarly, the velocity prediction loss corresponds to the OT-flow matching loss:

$$\mathbb{E}_{x \sim \pi(x), \epsilon \sim \mathcal{N}(0, \mathbf{I}_d)}[\|V_\phi((1-t)x + t\epsilon, t) - (\epsilon - x)\|^2] \tag{22}$$

and flow model corresponds to the velocity predictor ODE. Therefore, **diffusion models and OT-flow matching are both HGFs with the zero force field** - the difference lies in a coupled construction of the initial distribution (see Figure 2). The coupled construction allows OT-flow matching to have straighter probability paths, leading to improved generation quality for the same number of steps [31].

# 7 Oscillation HGFs

So far, we studied optimal velocity predictors $V_\phi$ for two extreme cases: either $F_\theta = 0$ or $F_\theta = \nabla \log \pi$. Finally, we want to investigate a different choice of $F_\theta$ to construct HGFs. Here, we study *Oscillation HGFs* that correspond to a natural extension. In Appendix I, we give another example that we coin *Reflection HGFs*.

A simple design of a force field is to use the physical model of a harmonic oscillator, i.e. to set $F_\theta(x) = -\alpha^2 x$ with $\Pi = \pi_{BG}$ and $\alpha > 0$. The flow of the ODE then becomes:

$$(x_t, v_t) = \left(\cos(\alpha t)x + \tfrac{1}{\alpha}\sin(\alpha t)v, -\alpha\sin(\alpha t)x + \cos(\alpha t)v\right) \tag{23}$$

I.e. condition (C1) is fulfilled as we can simply compute the ODE analytically. Setting $T = \pi/(2\alpha)$, it holds that $(x_t, v_t) = (v, -\alpha x)$. In particular, $\pi_T = \mathcal{N}(0, \mathbf{I}_d/\alpha^2)$ - condition (C1) is easily fulfilled. Therefore, the above choice gives us a natural generative model based on harmonic oscillators that we coin *Oscillation HGFs*. To summarize, they have the following simple training objective:

$$\mathbb{E}_{x \sim \pi, v \sim \mathcal{N}(0, \mathbf{I}_d)}[\|V_\phi(\cos(\alpha t)x + \frac{\sin(\alpha t)}{\alpha}v, t) - [-\alpha\sin(\alpha t)x + \cos(\alpha t)v]\|^2] \tag{24}$$

A natural choice for $\alpha$ is to set $\alpha = \sqrt{d/\mathbb{E}_{x \sim \pi}[\|x\|^2]}$. With this, the scale of the $n$-th derivative (including $n = 0$) of the inputs and outputs in the training objective is constant in time (see Figure 2), i.e. for all $t = 0$:

$$\mathbb{E}_{x \sim \pi, v \sim \mathcal{N}(0, \mathbf{I}_d)}\left[\|\frac{d^n}{d^n t}x_t\|^2\right] = \alpha^{n-2}d, \quad \mathbb{E}_{x \sim \pi, v \sim \mathcal{N}(0, \mathbf{I}_d)}\left[\|\frac{d^n}{d^n t}v_t\|^2\right] = \alpha^n d \tag{25}$$

In the context of critically-damped Langevin diffusion [13], it was already observed that a constant scale in velocity space leads to improved training and better performance. Here, we extend this idea of a constant scale from the velocity to the $n$-th derivative.

# 8 Related Work

**Assessing and training energy-based models.** Stein's discrepancy [16] is a well-known measure to assess the quality of energy-based models based on Stein's identity [46]. Based on this metric,

[17] developed a method that is similar to ours where a critic is optimized to assess the quality of an energy-based model via Stein's discrepancy and jointly trained with the energy model via min-max optimization. [25] introduced score matching as a method by showing that the explicit score matching loss (see Equation (5)) can be implicitly trained if one computes the trace of Hessian of the energy function - an expensive step. To expedite this, [34] introduced curvature propagation for an unbiased Hessian estimate, while [43] used Hutchinson's Trick to estimate the trace. In practice, both methods suffer from high variance due to their underlying Monte Carlo estimators.

**Flow matching and Stochastic interpolants.**    As already seen for a special case in Section 6, HGFs are strongly connected to Flow Matching [31] and Stochastic Interpolants [3]. They construct probability paths that fulfill the continuity equation by predicting derivatives of flows (i.e. velocities) in the same way how in this work, we predict velocities as marginals of an extended state space. The differences of these 3 works lie in the design perspective: HGFs consider 2nd-order ODEs in an extended state space $\mathbb{R}^d \times \mathbb{R}^d$ with a simple *initial velocity distribution* (here, $\mathcal{N}(0, \mathbf{I}_d)$), while FM considers 1st-order ODE paths in $\mathbb{R}^d$ converging to a simple *final location distribution*. Flow matching conditions on final states (usually at $t = 1$), while our framework conditions on the velocity of the current state (arbitrary $t$) and is centered around forces. This work arrives at the ideas of conditional velocity predictors via the search of properties that are conserved under Hamiltonian dynamics (see Theorem 1). We elucidate the mathematical connection in more detail in Appendix J.

**Generative models and Hamiltonian physics.**    V-Flows also consider augmenting the state space with velocities deriving an ELBO objective for CNFs [6]. [13] extend diffusion models to joint state-velocity samples that converge to a joint normal distribution. One difference is that we only need to run the backward equation in state space $\mathbb{R}^d$ as opposed to extended state-velocity space $\mathbb{R}^d \times \mathbb{R}^d$. Though rather unmotivated, Oscillation HGFs could, in principle, also be derived as an EDM model with preconditioning [26]. Finally, several works have, like this, explored generative models based on specific physical processes, e.g. Poisson flow generative models [50, 51]. A few works also combined Hamiltonian physics with deep learning. For example, [18] use conservation of energy as an implicit bias to learn networks for physical data. Conversely, deep learning was also used to accelerate HMC sampling [15], e.g. by training custom MCMC kernels [30] or correct for complex geometries via flows [22]. Very recently, score matching approaches were also designed to leverage existing force fields as part of a diffusion model that samples from an energy landscape [2, 12].

**Acceleration Generative Model (AGM).**    The AGM model [9] also uses constructions in phase space (joint position and velocity space) and 2nd order ODEs. While AGM focuses on learning the force field, our approach primarily focuses on learning the optimal velocity predictor. While we also consider optimizing the force field by minimizing the norm of the optimal velocity predictor, this happens in the "outer loop" of the maximization - the inner loop optimizes the optimal velocity predictor. Further, ATM focuses on bridging two desired distributions by posing a stochastic bridge problem in phase space. We do not consider the problem of bridging distributions. In contrast, our framework centers around energy preservation and divergence from that preservation (for optimal velocity predictors that are not zero). Specifically, we establish a connection to Hamiltonian physics and a property of the preservation of energy. This allows us to introduce a further bi-level optimization and the possibility of joint training for score matching.

# 9    Experiments

## 9.1    Hamiltonian Score Discrepancy

As we introduced Hamiltonian score discrepancy as a novel score-matching metric, we first empirically investigate our theoretical insights on Gaussian mixtures (see Figure 1). As one can see in Figure 3a, the Hamiltonian score discrepancy is highly correlated with the explicit score matching loss. Further, we can validate empirically that the Taylor approximation derived in Proposition 2 is pretty accurate for large $t$ (see Appendix I). Overall, these results indicate that the Hamiltonian score discrepancy is a natural metric to assess score approximations.

Further, we investigate whether explicitly minimizing the Hamiltonian score discrepancy leads to accurate score approximations. We jointly train velocity predictors and score networks as described in Section 4. As one can see visually in Figure 1, this approach can faithfully learn score vector fields. In addition, we investigate the signal-to-noise ratio for gradient estimation. As shown in Figure 3c, the gradient estimates of HSM have significantly lower variance compared to denoising

score matching at lower noise levels $\sigma$. The reason for that is that we allow for supervision across a full trajectory at locations for the same data points - effectively acting as data augmentation. Of course, this comes at the expense of simulating the Hamiltonian trajectories for $\sim 5$ steps.

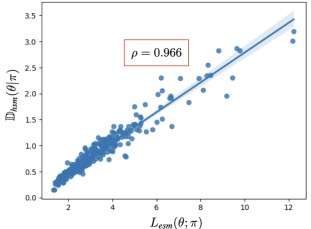
(a) ESM loss vs HSD for networks trained for 1 epoch

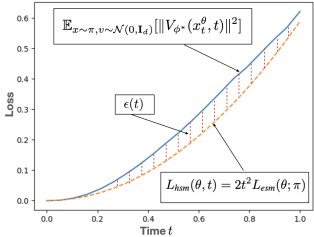
(b) Empirical HSD vs. Taylor approximation (see Proposition 2)

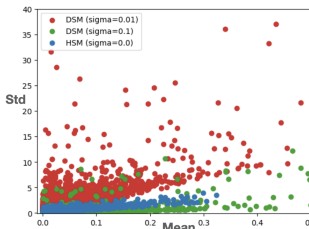
(c) Std vs absolute mean of derivative of param. of score network.

Figure 3: Empirical investigation of Hamiltonian score discrepancy (HSD). (a) The Taylor approximation is a good approximation. (b) Hamiltonian score discrepancy is strongly correlated with explicit score matching loss. (c) Signal-to-noise ratio is significantly better for HSM vs DSM for low $\sigma$.

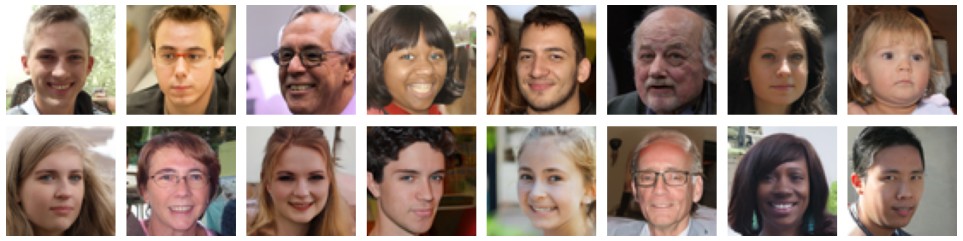

Figure 4: Image generation examples based on Oscillation HGFs for FFHQ.

## 9.2 HGF experiments - Image Generation

In the form of diffusion models and flow matching, HGFs have already been extensively optimized and achieved state-of-the-art results. Instead, we investigate whether also other non-zero force fields, specifically Oscillation HGFs, can lead to generative models of high quality. We focus on image generation benchmarks.

Specifically, we train a Oscillation HGF on CIFAR-10 unconditional and conditional. As two central benchmarks, we use the original SDE formulation of diffusion models [45] as well as the EDM framework [26], a highly-tuned optimization of diffusion models. Our hypothesis is that Oscillation HGFs should work well out-of-the-box, as the scale of their inputs and outputs stay around constant (*c.f.* Eq. 25). Therefore, we remove any preconditioning optimized for diffusion models (scaling of inputs and outputs and skip connections) [26] and train on the default DDPM architecture [21] (see details in Appendix L). Our results are encouraging: without hyperparameter tuning, Oscillations HGFs can sample high-quality images and surpass most previous methods (see Table 1) measured by Frechet Inception Distance (FID) [20]. While they still lack behind the EDM model, this difference might well be explained by the fact that

Table 1: Sample quality (FID) and number of function evaluation (NFE).

| METHOD | FID ↓ | NFE ↓ |
|---|---|---|
| *CIFAR-10 (unconditional)-32x32* | | |
| StyleGAN2-ADA [27] | 2.92 | 1 |
| DDPM [21] | 3.17 | 1000 |
| LSGM [47] | 2.10 | 147 |
| PFGM [50] | 2.48 | 104 |
| VE-SDE [45] | 3.77 | 35 |
| VP-SDE [45] | 3.01 | 35 |
| EDM [26] | 1.98 | 35 |
| FM-OT (BNS) [40] | 2.73 | 8 |
| Oscillation HGF (ours) | 2.12 | 35 |
| *CIFAR-10 (class conditional)-32x32* | | |
| VE-SDE [45] | 3.11 | 35 |
| VP-SDE [45] | 2.48 | 35 |
| EDM [26] | 1.79 | 35 |
| Oscillation HGF (ours) | 1.97 | 35 |
| *FFHQ (unconditional)-64x64* | | |
| VE-SDE [45] | 25.95 | 79 |
| VP-SDE [45] | 3.39 | 79 |
| EDM [26] | 2.39 | 79 |
| Oscillation HGF (ours) | 2.86 | 79 |

architectures and hyperparameters have been optimized for diffusion models over several works that are hard to replicate.

To investigate whether similar results can be achieved similar performance at higher resolutions, we perform another benchmark on the FFHQ dataset at 64x64 resolution. Here, our results are similar: Oscillation HGFs improve upon original diffusion models with a small performance margin to the EDM model. They can generate high-quality faces that appear realistic (see Figure 4).

## 10  Conclusion

Our work systematically elucidates the synergy between Hamiltonian dynamics, force fields, and generative models - extending and giving a new perspective on many known generative models. We believe that this opens up new avenues for applications of machine learning in physical sciences and dynamical systems. However, several limitations remain. Minimizing the Hamiltonian Score Discrepancy (HSD) via a default min-max algorithm is scalable but requires adversarial optimization. Future work can focus on adapting the HSD framework, e.g. to develop *denoising* Hamiltonian score matching that could allow for guaranteed convergence. For HGFs, the extended design space presented has only been explored for data without known force fields (image generation, here). Future work can focus on specific applications that require domain-specific force fields, e.g. for molecular data. Further adaptions might be required in such settings as such data often lie on manifolds. A further challenge is that HGFs not necessarily converge to a known distribution for more complex force fields. Therefore, we anticipate that future work will focus on adapting HGFs and related models to this challenge to design domain-specific models.

### Acknowledgments and Disclosure of Funding

This work was funded in part by GIST-MIT Research Collaboration grant (funded by GIST), the Machine Learning for Pharmaceutical Discovery and Synthesis (MLPDS) consortium, the DTRA Discovery of Medical Countermeasures Against New and Emerging (DOMANE) threats program, and the NSF Expeditions grant (award 1918839) Understanding the World Through Code.

P.H. would like to thank Gabriele Corso, Ziming Liu, and Timur Garipov for helpful discussions during early stages of the work.

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

# A    Score matching methods

To make score matching tractable, one can express the explicit score matching loss via [25]

$$L_{\text{ism}}(\theta; \pi) = \mathbb{E}_{x \sim \pi} \left[ \nabla \cdot F_\theta(x) + \frac{1}{2} \|F_\theta(x)\|^2 \right] = L_{\text{esm}}(\theta; \pi) + C, \tag{26}$$

where $\nabla \cdot F_\theta$ is the divergence of the vector field and the constant $C$ is independent of $\theta$. While this loss has been widely used [29], this objective is hard to optimize with neural networks as the divergence requires to backpropagate $d$ times. Still, the divergence can be approximated via Hutchinson's trick [24] leading to *sliced score-matching* [43].

# B    Hamiltonian ODE: Conservation of energy and volume

In this section, we prove the fundamental properties of the flow $\varphi_t$ of the Hamiltonian ODE (see Equation (2)). As these properties are used throughout this work and usually presented in the context of the physics literature, we include the proofs here for completeness, solely expressing it in the language of probability theory. Throughout this section, let $J \in \mathbb{R}^{2d \times 2d}$ be the matrix defined by:

$$J = \begin{pmatrix} 0 & \mathbf{1}_d \\ -\mathbf{1}_d & 0 \end{pmatrix}$$

## B.1    Preservation of energy.

**Proposition 4.** *For all $t \in \mathbb{R}$ it holds that $H \circ \varphi_t = H$*

*Proof.* We follow [5, theorem 2.2]. For all $z \in \mathbb{R}^{2d}$ one has $\langle z, Jz \rangle = 0$. Hence

$$\frac{d}{dt} H(\varphi_t(z)) = \langle \nabla H(\varphi_t(z)), \frac{d}{dt}\varphi_t(z) \rangle = \langle \nabla H(\varphi_t(z)), J \nabla H(\varphi_t(z)) \rangle = 0$$

which implies that $H(\varphi_t(z)) = H(\varphi_0(z)) = H(z)$. $\qquad\square$

## B.2    Preservation of volume.

**Proposition 5** (Symplectic property). *For all $z = (x, v) \in \mathbb{R}^{2d}$ and $t \in \mathbb{R}$ the Jacobian $D\varphi_t(z)$ satisfies the following equation:*

$$D\varphi_t(z)^T J^T D\varphi_t(z) = J^T \tag{27}$$

*In particular, $|det D\varphi_t| \equiv 1$, i.e. the $\varphi_t$ is volume-preserving.*

*Proof.* Here, we follow [33, proposition 2.6.2]. First of all, it can be easily seen that Equation (27) is equivalent to the statement that the bilinear form $\beta(u, v) = \langle u, J^T v \rangle$ on $\mathbb{R}^{2d}$ fulfils

$$\beta(u, v) = \beta(D\varphi_t(z)u, D\varphi_t(z)v) \quad \forall z, u, v \in \mathbb{R}^{2d} \tag{28}$$

By taking derivatives of the right-hand side and afterwards using the identities $\beta(Jx, y) = \langle J^2 x, y \rangle = -\langle x, y \rangle$ and $\beta(x, Jy) = \langle x, J^T J y \rangle = \langle x, y \rangle$, one gets:

$$\frac{d}{dt}\beta(D\varphi_t(z)u, D\varphi_t(z)v) = \beta(\frac{d}{dt}D\varphi_t(z)u, D\varphi_t(z)v) + \beta(D\varphi_t(z)u, \frac{d}{dt}D\varphi_t(z)v)$$

$$= \beta(D\frac{d}{dt}\varphi_t(z)u, D\varphi_t(z)v) + \beta(D\varphi_t(z)u, D\frac{d}{dt}\varphi_t(z)v)$$

$$= \beta(J\nabla^2 H(\varphi_t(z))D\varphi_t(z)u, D\varphi_t(z)v) + \beta(D\varphi_t(z)u, J\nabla^2 H(\varphi_t(z))D\varphi_t(z)v)$$

$$= -\langle \nabla^2 H(\varphi_t(z))D\varphi_t(z)u, D\varphi_t(z)v \rangle + \langle D\varphi_t(z)u, \nabla^2 H(\varphi_t(z))D\varphi_t(z)v \rangle$$

$$= 0$$

where in the last step I have used that the Hessian is symmetric. One can conclude

$$\beta(D\varphi_t(z)u, D\varphi_t(z)v) = \beta(D\varphi_0(z)u, D\varphi_0(z)v) = \beta(u, v)$$

Since $|\det J| = 1$, one immediately gets $|\det D\varphi_t| \equiv 1$. $\qquad\square$

# C   Proof of Theorem 1

## C.1   (1)⇒(2): Score preserves Boltzmann-Gibbs distribution

This section gives proof that the Hamiltonian ODE defined with the score preserves the Boltzmann-Gibbs distribution. This implication is well-known, and the proof is included for completeness following [5].

For all $t \in \mathbb{R}$ and Borel sets $D$, it holds by Propositions 4 and 5 and a change of variables:

$$\pi_{BG}(\varphi_t(D)) = Z^{-1} \int \mathbf{1}_{\varphi_t(D)} \exp(-H) dz \tag{29}$$

$$= Z^{-1} \int \mathbf{1}_{\varphi_t(D)} \circ \varphi_t \exp(-H \circ \varphi_t) |\det D\varphi_t| dz \tag{30}$$

$$= Z^{-1} \int \mathbf{1}_D \exp(-H)| dz \tag{31}$$

$$= \pi_{BG}(D) \tag{32}$$

Note that the symplectic property is crucial, e.g. consider the simple pendulum ($U(x) = \frac{1}{2}x^2$) and $g(z) = (\|z\|, 0)$. This function fulfills $\pi_{BG}(g(z)) = \pi_{BG}(z)$ but it does not leave the distribution invariant (it is not symplectic, either).

## C.2   (2)⇒(1): Invariance under Boltzmann-Gibbs uniquely defines score

We write $\varphi_t(z) = (x_t^{\nabla U}, v_t^{\nabla U})$ for the solution with force field $-\nabla U$. Let's pick an arbitrary sufficiently regular test function $f : \mathbb{R}^{2d} \to \mathbb{R}$ and let's define

$$F_\theta(t) = \mathbb{E}_{z \sim \pi_{BG}}[f(\varphi_t^\theta(z))]$$
$$F_{\nabla U}(t) = \mathbb{E}_{z \sim \pi_{BG}}[f(\varphi_t(z))]$$

As the Hamiltonian dynamics with force network $-\nabla U$ preserve the Boltzmann-Gibbs distribution, $F_{\nabla U}$ must be a constant function, i.e. it derivative is zero. Therefore, we can compute:

$$0 = \frac{d}{dt} F_{\nabla U}(t)$$

$$= \frac{d}{dt} \mathbb{E}_{z \sim \pi_{BG}}[f(\varphi_t(z))]$$

$$= \mathbb{E}_{z \sim \pi_{BG}} \left[ \frac{d}{dt} f(\varphi_t(z)) \right]$$

$$= \mathbb{E}_{z \sim \pi_{BG}} \left[ \nabla_z f(\varphi_t(z)^T \frac{d}{dt} \varphi_t(z) \right]$$

$$= \mathbb{E}_{z \sim \pi_{BG}} \left[ \nabla_z f(\varphi_t(z))^T \begin{pmatrix} v_t^{\nabla U}(z) \\ -\nabla U(x_t^{\nabla U}(z)) \end{pmatrix} \right]$$

I.e. at time $t = 0$:

$$0 = \frac{d}{dt} F_{\nabla U}(t)_{|t=0} = \mathbb{E}_{z=(x,v) \sim \pi_{BG}} \left[ \nabla_z f(z)^T \begin{pmatrix} v \\ -\nabla U(x) \end{pmatrix} \right] \tag{33}$$

As $\varphi_t^\theta$ also preserves the Boltzmann-Gibbs distribution, we can follow the same computation to also get that:

$$0 = \frac{d}{dt} F_\theta(t)_{|t=0} = \mathbb{E}_{z=(x,v) \sim \pi_{BG}} \left[ \nabla_z f(z)^T \begin{pmatrix} v \\ F_\theta(x) \end{pmatrix} \right] \tag{34}$$

Substracting Equation (33) from Equation (34), we get that:

$$0 = \mathbb{E}_{z=(x,v) \sim \pi_{BG}} \left[ \nabla_z f(z)^T \begin{pmatrix} 0 \\ F_\theta(x) + \nabla U(x) \end{pmatrix} \right] \tag{35}$$

Let's set $f(z) = f(x, v) = v^T(F_\theta(x) + \nabla U(x))$. Then

$$\nabla_z f(x, v) = \begin{pmatrix} v^T(DF_\theta(x) + \nabla^2 U(x)) \\ F_\theta(x) + \nabla U(x) \end{pmatrix} \tag{36}$$

where $DF_\theta$ denotes the Jacobian and $\nabla^2 U$ the Hessian. Then inserting Equation (36) into Equation (35) we get that:

$$\begin{aligned}
0 =& \mathbb{E}_{z=(x,v)\sim\pi_{BG}} \left[ \begin{pmatrix} v^T(DF_\theta(x) + \nabla^2 U(x)) \\ F_\theta(x) + \nabla U(x) \end{pmatrix}^T \begin{pmatrix} 0 \\ F_\theta(x) + \nabla U(x) \end{pmatrix} \right] \\
=& \mathbb{E}_{z=(x,v)\sim\pi_{BG}} \left[ \|F_\theta(x) + \nabla U(x)\|^2 \right] \\
=& \mathbb{E}_{x\sim\pi} \left[ \|F_\theta(x) - [-\nabla U(x)]\|^2 \right]
\end{aligned}$$

This implies that $F_\theta(x) = -\nabla U(x) = \nabla \log \pi$ for $\pi$-almost every $x$.

## C.3 (3) $\Leftrightarrow$ (1)

Finally, we show that condition (3) is equivalent to condition (1). Note that if $(x_t^\theta, v_t^\theta) \sim \pi_{BG} = \pi \otimes \mathcal{N}(0, \mathbf{I}_d)$, then $v_t^\theta$ is independent of $x_t^\theta$ and it holds that $\mathbb{E}[v_t^\theta | x_t^\theta] = 0$. Therefore, condition (2) trivially implies condition (3). As we have already have shown that condition (2) is equivalent to condition (1), it is sufficient to prove that (3) implies (1). The proof is similar to the proof for (2) $\Rightarrow$ (1).

Let's assume that the following condition holds for $z \sim \pi_{BG}$

$$0 = \mathbb{E}[v_t^\theta | x_t^\theta = x] \quad \text{for all } x \tag{37}$$

where for brevity we write $(x_t^\theta, v_t^\theta) = \varphi_t^\theta(z)$. Then for $f(z) = f(x, v) = v^T(F_\theta(x) + \nabla U(x))$ as selected above, we can equally derive:

$$\begin{aligned}
F_\theta(t) =& \mathbb{E}_{z\sim\pi_{BG}}[f(\varphi_t^\theta(z))] \\
=& \mathbb{E}[f(x_t^\theta, v_t^\theta)] \\
=& \mathbb{E}[(v_t^\theta)^T(F_\theta(x_t^\theta) + \nabla U(x_t^\theta))] \\
=& \mathbb{E}[\mathbb{E}[(v_t^\theta)^T(F_\theta(x) + \nabla U(x))|x = x_t^\theta]] \\
=& \mathbb{E}[\mathbb{E}[v_t^\theta | x_t^\theta]^T(F_\theta(x_t^\theta) + \nabla U(x_t^\theta))] \\
=& \mathbb{E}\left[0^T(F_\theta(x_t^\theta) - \nabla U(x_t^\theta))\right] \\
=& \mathbb{E}[0] \\
=& 0
\end{aligned}$$

In the same way as above, we can now deduce that $\frac{d}{dt}F_\theta(t)_{|t=0} = 0$. Similarly to above, we can complete the proof by using Equation (36):

$$\begin{aligned}
0 =& \frac{d}{dt}\mathbb{E}_{z\sim\pi_{BG}}[f(\varphi_t^\theta(z))] - \frac{d}{dt}\mathbb{E}_{z\sim\pi_{BG}}[f(\varphi_t(z))] \\
=& \mathbb{E}_{z=(x,v)\sim\pi_{BG}} \left[ \nabla_z f(z)^T \begin{pmatrix} 0 \\ F_\theta(x) + \nabla U(x) \end{pmatrix} \right] \\
=& \mathbb{E}_{x\sim\pi} \left[ \|F_\theta(x) - [-\nabla U(x)]\|^2 \right]
\end{aligned}$$

Again, we can deduce that $F_\theta(x) = -\nabla U(x) = \nabla \log \pi(x)$ for $\pi$-almost every $x$.

## D  Proof of Equation (12)

We formally proof the identity in Equation (12). We will use the following well-known characterization of the expectation:

**Lemma 1.** *Let $X \in \mathbb{R}^d$ be a random variable. Then:*

$$\mathbb{E}[X] = \arg\min_{m\in\mathbb{R}^d} \mathbb{E}[\|X - m\|^2] \tag{38}$$

*Proof.* Let $c = \mathbb{E}[X] \in \mathbb{R}^d$, then

$$
\begin{aligned}
\mathbb{E}[\|X - m\|^2] &= \mathbb{E}[\|X - c + c - m\|^2] \\
&= \mathbb{E}[\|X - c\|^2 + 2(X - c)^T(X - m) + \|c - m\|^2] \\
&= \mathbb{E}[\|X - c\|^2] + 2(\mathbb{E}[X] - c)^T(X - m)] + \|c - m\|^2 \\
&= \mathbb{E}[\|X - c\|^2] + \|c - m\|^2 \\
&\geq \mathbb{E}[\|X - c\|^2]
\end{aligned}
$$

This implies the statement. $\qquad\square$

Therefore, we can apply Lemma 1 conditionally on $x_t^\theta$ to see that

$$
V_{\phi^*}(x, t) = \mathbb{E}[v_t^\theta | x_t^\theta = x]
$$

# E    Proof of Proposition 1

We can derive:

$$
\begin{aligned}
L_{\text{hsm}}(\phi|\theta, t) &= \mathbb{E}_{z \sim \pi_{BG}} \left[ -2V_\phi(x_t^\theta, t)^T v_t^\theta + \|V_\phi(x_t^\theta, t)\|^2 \right] &(39) \\
&= \mathbb{E}_{z \sim \pi_{BG}} \left[ \|v_t^\theta\|^2 - 2V_\phi(x_t^\theta, t)^T v_t^\theta + \|V_\phi(x_t^\theta, t)\|^2 \right] - \mathbb{E}_{z \sim \pi_{BG}} \left[ \|v_t^\theta\|^2 \right] &(40) \\
&= \mathbb{E}_{z \sim \pi_{BG}} \left[ \|V_\phi(x_t^\theta, t) - v_t^\theta\|^2 \right] - \mathbb{E}_{z \sim \pi_{BG}} \left[ \|v_t^\theta\|^2 \right] &(41) \\
& &(42)
\end{aligned}
$$

As $\mathbb{E}_{z \sim \pi_{BG}} \left[ \|v_t^\theta\|^2 \right]$ is a constant in $\phi$, we can see that

$$
\arg\min_\phi L_{\text{hsm}}(\phi|\theta, t) = \arg\min_\phi \mathbb{E}_{z \sim \pi_{BG}}[\|V_\phi(x_t^\theta, t) - v_t^\theta\|^2]
$$

Hence, the Hamiltonian velocity predictor is the minimizer of the above objective. Inserting Equation (12) into Equation (39), we get

$$
\begin{aligned}
\max_\phi L_{\text{hsm}}(\phi|\theta, t) &= L_{\text{hsm}}(\phi^*|\theta, t) \\
\\
&= \mathbb{E}_{z \sim \pi_{BG}} \left[ 2\mathbb{E}[v_t^\theta|x_t^\theta]^T v_t^\theta - \|\mathbb{E}[v_t^\theta|x_t^\theta]\|^2 \right] \\
&= \mathbb{E}_{z \sim \pi_{BG}} \left[ 2\mathbb{E}[\mathbb{E}[v_t^\theta|x_t^\theta]^T v_t^\theta|x_t^\theta] - \|\mathbb{E}[v_t^\theta|x_t^\theta]\|^2 \right] \\
&= \mathbb{E}_{z \sim \pi_{BG}} \left[ 2\mathbb{E}[v_t^\theta|x_t^\theta]^T \mathbb{E}[v_t^\theta|x_t^\theta] - \|\mathbb{E}[v_t^\theta|x_t^\theta]\|^2 \right] \\
&= \mathbb{E}_{z \sim \pi_{BG}} \left[ 2\|\mathbb{E}[v_t^\theta|x_t^\theta]\|^2 - \|\mathbb{E}[v_t^\theta|x_t^\theta]\|^2 \right] \\
&= \mathbb{E}_{z \sim \pi_{BG}} \left[ \|\mathbb{E}[v_t^\theta|x_t^\theta]\|^2 \right]
\end{aligned}
$$

where have used that the conditional expectation $\mathbb{E}[v_t^\theta|x_t^\theta]$ is a constant conditioned on $x_t^\theta$. This finishes the proof.

# F    Proof of Theorem 2

As $\lambda$ is a distribution with full support over $[0, T)$, it holds that

$$
\mathbb{D}_{\text{hsm}}(\theta|\pi) = \mathbb{E}_{t \sim \lambda, z \sim \pi_{BG}} \left[ \|\mathbb{E}[v_t^\theta|x_t^\theta]\|^2 \right] = 0
$$

if and only if for very $0 \leq t < T$ (up to measure zero)

$$
\mathbb{E}[v_t^\theta|x_t^\theta] = 0
$$

By Theorem 1, this is equivalent to $F_\theta = \nabla \log \pi$. Hence, $\mathbb{D}_{\text{hsm}}(\theta|\pi) = 0$ if and only if $F_\theta = \nabla \log \pi$.

**Remark.** Technically speaking, we maximized $V_\phi$ for a fixed $t$ in Proposition 1. However, we remark that maximizing it across $t$ leads to the same result for all $0 \leq t < T$ under reasonable regularity conditions. More specifically, assuming that $\pi$ is a smooth density with full support, the map $(x, t) \mapsto \mathbb{E}[v_t^\theta|x_t^\theta = x]$ is continuous in $t$ and $x$. Therefore, as long as $(V_\phi)_{\phi \in I}$ covers all continuous function in $t$ and $x$, the result above is the same.

# G  Proof of Proposition 2

The proof relies on a Taylor approximation of $L_{\text{hsm}}(\theta, t)$ around $t = 0$. To finish the proof, we have to show the following three equations:

$$\mathbb{D}_{\text{hsm}}(\theta|t, \pi)_{|t=0} = 0 \tag{43}$$

$$\frac{d}{dt}\mathbb{D}_{\text{hsm}}(\theta|t, \pi)_{|t=0} = 0 \tag{44}$$

$$\frac{d^2}{d^2t}\mathbb{D}_{\text{hsm}}(\theta|t, \pi)_{|t=0} = 4L_{\text{esm}}(\theta; \pi) = 2\mathbb{E}_{x\sim\pi}[\|F_\theta(x) - \nabla\log\pi(x)\|^2] \tag{45}$$

**Proof of Equation (43)**  Note that at time $t = 0$, it holds that $v_t^\theta = v \sim \mathcal{N}(0, \mathbf{I}_d)$ and $x_t^\theta = x \sim \pi$. Therefore,

$$\mathbb{E}[v_t^\theta|x_t^\theta]_{|t=0} = \mathbb{E}_{x\sim\pi, v\sim\mathcal{N}(0,\mathbf{I}_d)}[v|x] = \mathbb{E}_{v\sim\mathcal{N}(0,\mathbf{I}_d)}[v] = 0$$

Therefore, by Proposition 1

$$\mathbb{D}_{\text{hsm}}(\theta|0, \theta) = \mathbb{E}[\|\mathbb{E}[v_t^\theta|x_t^\theta]_{|t=0}\|^2] = \mathbb{E}[\|0\|^2] = 0$$

**Proof of Equation (44)**  We can compute:

$$\frac{d}{dt}\mathbb{D}_{\text{hsm}}(\theta|t, \pi) = \mathbb{E}[\frac{d}{dt}\|\mathbb{E}[v_t^\theta|x_t^\theta]\|^2] = 2\mathbb{E}[\mathbb{E}[v_t^\theta|x_t^\theta]^T\frac{d}{dt}\mathbb{E}[v_t^\theta|x_t^\theta]]$$

So at time $t = 0$:

$$\frac{d}{dt}\mathbb{D}_{\text{hsm}}(\theta|t, \pi)_{|t=0} = 2\mathbb{E}[\mathbb{E}[v|x]^T\frac{d}{dt}\mathbb{E}[v_t^\theta|x_t^\theta]_{|t=0}] = 2\mathbb{E}[0^T\frac{d}{dt}\mathbb{E}[v_t^\theta|x_t^\theta]_{|t=0}] = 0$$

**Proof of Equation (45)**  Let's take the second derivative:

$$\frac{d^2}{d^2t}\mathbb{D}_{\text{hsm}}(\theta|t, \pi) = 2\frac{d}{dt}\mathbb{E}[\mathbb{E}[v_t^\theta|x_t^\theta]^T\frac{d}{dt}\mathbb{E}[v_t^\theta|x_t^\theta]]$$

$$= 2\mathbb{E}[\mathbb{E}[v_t^\theta|x_t^\theta]^T\frac{d^2}{d^2t}\mathbb{E}[v_t^\theta|x_t^\theta]] + 2\mathbb{E}[\|\frac{d}{dt}\mathbb{E}[v_t^\theta|x_t^\theta]\|^2]$$

by the product rule. And at time $t = 0$:

$$\frac{d^2}{d^2t}\mathbb{D}_{\text{hsm}}(\theta|t, \pi)_{|t=0} = 2\mathbb{E}[\mathbb{E}[0^T\frac{d^2}{d^2t}\mathbb{E}[v_t^\theta|x_t^\theta]] + 2\mathbb{E}[\|\frac{d}{dt}\mathbb{E}[v_t^\theta|x_t^\theta]\|^2] = 2\mathbb{E}[\|\frac{d}{dt}\mathbb{E}[v_t^\theta|x_t^\theta]_{|t=0}\|^2] \tag{46}$$

Let $\pi_t^\theta : \mathbb{R}^d \times \mathbb{R}^d \to \mathbb{R}$ be the density of $(x_t^\theta, v_t^\theta)$ and compute:

$$\mathbb{E}[v_t^\theta|x_t^\theta = x] = \int v\pi_t^\theta(v|x)dv = \int v\frac{\pi_t^\theta(x, v)}{\pi_t^\theta(x)}dv \tag{47}$$

$$\frac{d}{dt}\mathbb{E}[v_t^\theta|x_t^\theta = x] = \int v\frac{d}{dt}\frac{\pi_t^\theta(x, v)}{\pi_t^\theta(x)}dv \tag{48}$$

$$= \int v\frac{\pi_t^\theta(x)\frac{d}{dt}\pi_t^\theta(x, v) - \pi_t^\theta(x, v)\frac{d}{dt}\pi_t^\theta(x)}{(\pi_t^\theta(x))^2}dv \tag{49}$$

Let $G(x, v) = (v, F_\theta(x))^T$ be the Hamiltonian vector field. By the deterministic Fokker-Planck equation [37], we can derive that:

$$\frac{d}{dt}\pi_t^\theta = -\nabla \cdot [\pi_t^\theta G] = -G^T\nabla\pi_t^\theta - \pi_t^\theta\nabla G$$

Note that the Jacobian of $F$ is given by

$$DG(x, v) = \begin{pmatrix} 0 & \mathbf{I}_d \\ DF_\theta(x) & 0 \end{pmatrix}$$

In particular, $G$ is divergence-free, i.e. $\nabla \cdot G = \mathrm{tr}(DG) = 0$. Therefore,

$$\frac{d}{dt}\pi_t^\theta = - G^T \nabla \pi_t^\theta$$

and for particular $x, v \in \mathbb{R}^d$:

$$\frac{d}{dt}\pi_t^\theta(x,v)_{|t=0} = - \begin{pmatrix} v \\ F_\theta(x) \end{pmatrix}^T \nabla \pi_0^\theta \tag{50}$$

As $\pi_0^\theta = \pi_{BG}$ by construction, we can derive that

$$
\begin{aligned}
\nabla \pi_0^\theta(x,v) &= \nabla \pi_{BG}(x,v) \\
&= \frac{1}{Z}\nabla[\exp(-U(x) - \frac{1}{2}\|v\|^2)] \\
&= -\frac{1}{Z}\begin{pmatrix} \nabla U(x) \\ v \end{pmatrix}\exp(-U(x) - \frac{1}{2}\|v\|^2) \\
&= -\begin{pmatrix} \nabla U(x) \\ v \end{pmatrix}\pi_{BG}(x,v)
\end{aligned}
$$

Inserting this into Equation (50), we get

$$
\begin{aligned}
\frac{d}{dt}\pi_t^\theta(x,v) &= \begin{pmatrix} v \\ F_\theta(x) \end{pmatrix}^T \begin{pmatrix} \nabla U(x) \\ v \end{pmatrix}\pi_{BG}(x,v) \\
&= v^T(F_\theta(x) + \nabla U(x))\pi_{BG}(x,v)
\end{aligned}
$$

And hence,

$$
\begin{aligned}
\frac{d}{dt}\pi_t^\theta(x) &= \int \frac{d}{dt}\pi_t^\theta(x,v)dv \\
&= \int v^T(F_\theta(x) + \nabla U(x))\pi_{BG}(x,v)dv \\
&= \left[\int (F_\theta(x) + \nabla U(x))\pi(x)dv\right]^T \left[\int v\mathcal{N}(v;0,\mathbf{I}_d)dv\right] \\
&= 0
\end{aligned}
$$

We can insert these identities into Equation (49) to get:

$$
\begin{aligned}
\frac{d}{dt}\mathbb{E}[v_t^\theta|x_t^\theta = x]_{|t=0} &= \int v\frac{\pi_t^\theta(x)\frac{d}{dt}\pi_t^\theta(x,v) - \pi_t^\theta(x,v)\frac{d}{dt}\pi_t^\theta(x)}{(\pi_t^\theta(x))^2}dv_{|t=0} \\
&= \int v\frac{\pi(x)v^T(F_\theta(x) + \nabla U(x))\pi_{BG}(x,v) - 0}{\pi(x)^2}dv \\
&= \int v\frac{v^T(F_\theta(x) + \nabla U(x))\pi(x)^2}{\pi(x)^2}\mathcal{N}(v;0\mathbf{I}_d)dv \\
&= \left[\int vv^T\mathcal{N}(v;0\mathbf{I}_d)dv\right](F_\theta(x) + \nabla U(x)) \\
&= \mathbf{I}_d(F_\theta(x) + \nabla U(x)) \\
&= F_\theta(x) + \nabla U(x)
\end{aligned}
$$

Combining this with Equation (46), we get that:

$$
\begin{aligned}
\frac{d^2}{d^2t}L(\theta,t)_{|t=0} &= 2\mathbb{E}[\|\frac{d}{dt}\mathbb{E}[v_t^\theta|x_t^\theta]_{|t=0}\|^2] \tag{51} \\
&= 2\mathbb{E}[\|F_\theta(x) + \nabla U(x)\|^2] \tag{52} \\
&= 2\mathbb{E}[\|F_\theta(x) - \nabla \log \pi(x)\|^2] \tag{53} \\
&= 4L_{\mathrm{esm}}(\theta;\pi) \tag{54}
\end{aligned}
$$

**Taylor approximation** Finally, we can combine the above derivations to get a Taylor approximation of $L_{\text{hsm}}(\theta, t)$ around $t = 0$, i.e. for $\epsilon : \mathbb{R} \to \mathbb{R}$ with $\lim_{t \to 0} \frac{1}{t^2}|\epsilon(t)| = 0$ we get

$$
\begin{aligned}
\mathbb{D}_{\text{hsm}}(\theta|t, \theta) &= \mathbb{D}_{\text{hsm}}(\theta|0, \theta) + t\frac{d}{dt}\mathbb{D}_{\text{hsm}}(\theta|t, \theta)_{|t=0} + \frac{1}{2}t^2\mathbb{D}_{\text{hsm}}(\theta|t, \theta)_{|t=0} + \epsilon(t) \\
&= \frac{1}{2}t^2 4 L_{\text{esm}}(\theta; \pi) + \epsilon(t) \\
&= 2t^2 L_{\text{esm}}(\theta; \pi) + \epsilon(t)
\end{aligned}
$$

This finishes the proof.

# H   Proof of Proposition 3

Again, let's consider a probability distribution $\pi : \mathbb{R}^d \to \mathbb{R}$ and the ODE

$$
(\frac{d}{dt}x(t), \frac{d}{dt}v(t))^T = (v(t), F_\theta(x(t), t))
$$

where we know allow $F_\theta$ to be time-dependent. Let $(x_t, v_t)$ be a solution to the above with ODE with $(x_0, v_0) = (x, v) \sim \pi \otimes \mathcal{N}(0, \mathbf{I}_d)$. In addition, write $\Pi(x, v, t)$ for the distribution at time $t$ (i.e. $(x_t, v_t) \sim \Pi(\cdot, \cdot, t)$) and the **location marginal**

$$
\int \Pi(x, v, t)dv = \pi(x, t)
$$

Finally, we write

$$
V(x, t) = \mathbb{E}[v_t|x_t] = \int v\pi(v|x, t)dv = V_{\phi^*}(x, t)
$$

for the optimal velocity predictor.

**Deriving marginal ODE.** We now show that the evolution of the first marginal can be replicated by an ODE that only depends on $V$. By the Fokker-Planck equation, we can derive for $G(x, v) = (v, F_\theta(x))^T$:

$$
\begin{aligned}
\frac{\partial}{\partial t}\Pi(x, v, t) &= -\nabla_{x,v} \cdot [\Pi G](x, v, t) \\
&= -G(x, v, t)^T \nabla_{x,v}\Pi(x, v, t) - [\nabla_{x,v} \cdot G(x, v, t)]\Pi(x, v, t) \\
&= -\begin{pmatrix} v \\ F_\theta(x) \end{pmatrix}^T \nabla_{x,v}\Pi(x, v, t) \\
&= -[v^T \nabla_x \Pi(x, v, t) + F_\theta(x)^T \nabla_v \Pi(x, v, t)]
\end{aligned}
$$

where we used in the third equation that $\nabla_{x,v} \cdot G = 0$ (i.e. that $G$ is divergence-free). Therefore, we can derive that:

$$\frac{\partial}{\partial t}\pi(x,t) = \int \frac{\partial}{\partial t}\Pi(x,v,t)dv$$

$$= -\int [v^T \nabla_x \Pi(x,v,t) + F_\theta(x)^T \nabla_v \Pi(x,v,t)]dv$$

$$= -\int v^T \nabla_x \Pi(x,v,t)dv - F_\theta(x)^T \int \nabla_v \Pi(x,v,t)dv$$

$$= -\int v^T \nabla_x \Pi(x,v,t)dv - F_\theta(x)^T 0$$

$$= -\int v^T \nabla_x \Pi(x,v,t)dv$$

$$= -\nabla_x \cdot \int v\Pi(x,v,t)dv$$

$$= -\nabla_x \cdot [\Pi(x,t) \int v\frac{\Pi(x,v,t)}{\Pi(x,t)}dv]$$

$$= -\nabla_x \cdot [\Pi(x,t) \int v\Pi(v|x,t)dv]$$

$$= -\nabla_x \cdot [\Pi(x,t)V(x,t)]$$

In other words, the vector field $V(x,t)$ satisfies the continuity equation. Therefore, if we initialize $x_T \sim \pi(\cdot,T)$ and evolve the ODE

$$\frac{d}{dt}x(t) = V(x(t),t)$$

backwards until $t = 0$, we know that $x(0) \sim \pi(\cdot,0) = \pi$ - we sample from our data distribution.

# I  Reflection HGFs - A New Example of HGFs

We briefly show that HGFs can give us models other than Oscillation HGFs. For this, we introduce *Reflection HGFs* here and plot the result of training them in Figure 5 and Figure 6. The idea of the model is that particles can move freely in a box without collision with walls ("very strong forces") at the boundaries of the data domain making the particles bounce back (this can be made rigorous with von Neumann boundary conditions). With normally distributed velocities, the distribution of particles will converge towards a uniform distribution. Further, this model can be trained in a simulation-free manner. We trained this HGF model on a simple toy distribution (see figure 3 and figure 4 in the attached PDF). Such a HGF model is distinct from previous models and illustrates that HGFs are not restricted to only Oscillation HGFs, diffusion models, or flow matching.

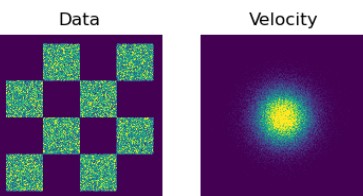

Figure 5: Data distribution (left) and velocity distribution (right) used for Reflection HGFs as initial distribution. With the above starting conditions, a reflection (="infinite force") at the boundaries of the domain is used to simulate trajectories forward (this can be computed in closed form in a simulation-free manner).

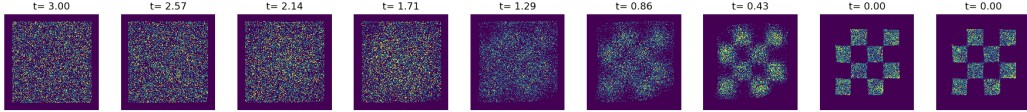

Figure 6: Illustration of sampling with trained Reflection HGFs. At time $t = 3.0$, the distribution is a uniform distribution (sampled by construction). By running the parameterized Hamiltonian ODE backwards in time, we recover the data distribution (see Figure 5).

## J  Connection between Flow Matching and HGFs

Let $A : \mathbb{R}^d \times \mathbb{R} \to \mathbb{R}^d$ be a time-dependent vector field and $\psi$ the diffeomorphic flow defined by the ODE:

$$\frac{d}{dt}x(t) = A(x(t), t)$$
$$x(0) = x$$

i.e. $t \mapsto \psi_t(x)$ is a solution to the above ODE. With this, we get:

$$\frac{d^2}{d^2t}x(t) = \frac{d}{dt}\left[\frac{d}{dt}x(t)\right]$$
$$= \frac{d}{dt}A(x(t), t)$$
$$= D_x A(x(t), t)\dot{x}(t) + \frac{\partial}{\partial t}A(x(t), t)$$
$$= D_x A(x(t), t)A(x(t), t) + \frac{\partial}{\partial t}A(x(t), t)$$

Therefore, if we define the force field,

$$F(x, t) = D_x A(x, t)A(x, t) + \frac{\partial}{\partial t}A(x, t)$$

we can extend the state space to $(x, v)$ and consider the ODE:

$$(x(0), v(0)) = (x, A(x, 0))$$
$$(\frac{d}{dt}x(t), \frac{d}{dt}v(t)) = (v(t), F(x(t), t))$$

Then every solution $(x_t, v_t)$ to the above ODE is also a solution to the flow matching ODE and vice versa. The conditional velocity predictor loss looks as follows:

$$\mathbb{E}[\|V_\phi(\psi_t(x_t), t) - \frac{d}{dt}\psi_t(x_t)\|^2]$$

This is exactly the conditional flow matching loss (see equation (14) in [31]).

## K  Connection between EDM and Oscillation HGFs

In this section, we discuss the relation between HGF and EDM [26]. The EDM paper assumes the perturbation kernel is isotropic Gaussian with standard deviation $\sigma(t)$. Thus, the intermediate distribution $p_t(\cdot; \sigma(t)) = p_{data} * \mathcal{N}(\mathbf{0}, \sigma(t)\mathbf{I})$. If we further scale the original variable $x$ with $s(t)$ and consider $\tilde{y} = s(t)x$, [26] shows that the corresponding backward ODE of $\tilde{y}$ is as follows:

$$d\tilde{y} = [\dot{s}(t)\tilde{y}/s(t) - s(t)^2\dot{\sigma}(t)\sigma(t)\nabla_{\tilde{y}}\log p(\tilde{y}/s(t); \sigma(t))]dt \tag{55}$$

We will show that, the minimizer of the objective of the Oscillation HFG, *i.e.,* $\mathbb{E}_{y\sim\pi, v\sim\mathcal{N}(0,\mathbf{I}_d)}[\|V_\phi(\cos(t)y + \sin(t)v, t) - [-\sin(t)y + \cos(t)v]\|^2]$, equals to the drift term in Eq. 55, when setting $s(t) = \cos(t)$ and $\sigma(t) = \tan(t)$.

Denote $\tilde{y} = \cos(t)y + \sin(t)v$, then the training objective can be rewritten as $\mathbb{E}_{y\sim\pi, v\sim\mathcal{N}(0,\mathbf{I}_d)}[\|V_\phi(\tilde{y},t) - [-\frac{y}{\sin(t)} + \frac{\tilde{y}}{\tan(y)}]\|^2]$. The minimizer of the training objective is

$$V_\phi^*(\tilde{y},t) = \mathbb{E}_{y|\tilde{y}}\left[-\frac{y}{\sin(t)}\right] + \frac{\tilde{y}}{\tan(y)} \tag{56}$$

On the other hand, we can re-express the score function in Eq. 55 as

$$\nabla_{\tilde{y}}\log p(\tilde{y}/\cos(t);\tan(t)) = \nabla_{\frac{\tilde{y}}{\cos(t)}}\log p(\tilde{y}/\cos(t);\tan(t))\frac{1}{\cos(t)}$$
$$= \frac{\mathbb{E}_{y|\tilde{y}}[y] - \tilde{y}/\cos(t)}{\tan^2(t)}\frac{1}{\cos(t)} \tag{57}$$

Plug Eq. 57 into the backward ODE (Eq. 55), we have:

$$\mathrm{d}\tilde{y} = [\dot{s}(t)\tilde{y}/s(t) - s(t)^2\dot{\sigma}(t)\sigma(t)\nabla_{\tilde{y}}\log p(\tilde{y}/s(t);\sigma(t))]\mathrm{d}t$$
$$= \left[-\tan(t)\tilde{y} - \tan(t)(\frac{\mathbb{E}_{y|\tilde{y}}[y] - \tilde{y}/\cos(t)}{\tan^2(t)}\frac{1}{\cos(t)})\right]\mathrm{d}t$$
$$= \left[\mathbb{E}_{y|\tilde{y}}\left[-\frac{y}{\sin(t)} + \frac{\tilde{y}}{\tan(y)}\right]\right]\mathrm{d}t \tag{58}$$

in which the drift term matches the optimal velocity predictor in Eq. 57. Hence, when picking the proper scaling factors, the backward ODE (Eq. 55) is equivalent to $\mathrm{d}\tilde{y} = V_\phi^*(\tilde{y},t)\mathrm{d}t$.

Recall that the EDM paper employs the simple scaling $s(t) = 1$ and $\sigma(t) = t$ in Eq. 55. Hence, to align with the time discretization $\{t_1, \ldots, t_n\}$ used in EDM during sampling, it suffices to set the time discretization in Oscillation HFG to $\{\arctan(t_1), \ldots, \arctan(t_n)\}$, to ensure that the score functions are evaluated on the same $\sigma$s.

**Remark.** Rescaling of the EDM ODE will necessarily lead to the same endpoint if $s(0) = 1$ - this is the case by construction. Similarly, changing the noise schedule will lead to the same ODE. However, mapping discretizations will *not* result in the same ODE. The reason for that is that in general

$$s'(t_{n+1})(t_{n+1} - t_n) \neq s(t_{t+1}) - s(t_n)$$
$$\sigma'(t_{n+1})(t_{n+1} - t_n) \neq \sigma(t_{t+1}) - \sigma(t_n)$$

## L  Details for Image Generation Benchmarks

In this section, we include more details about the training and sampling of Oscillation HGFs. All the experiments are run on 8 NVIDIA A100 GPUs. We used PyTorch as a library for automatic differentiation [38]. Our image processing pipeline follows [26]. We use the DDPM++ backbone [45, 21]. The preconditioning was removed. We set the reference batch size to 516 on CIFAR-10 and 256 on FFHQ. We train for 200 million images in total, corresponding to approximately 3000 epochs and $\sim 48$ hours of training time for CIFAR-10 and $\sim 96$ hours for FFHQ. As outlined in the experiments section, the hyperparameters and training procedure are the same as [26]: namely, we used the Adam optimizer with learning rate 0.001, exponential moving average (EMA) with momentum 0.5, data augmentation pipeline adapted from [28], dropout probability of 0.13, and FP32 precision. For sampling, we use the 2nd order Heun's sampler [26].

