# OpenReview forum: "Hamiltonian Score Matching and Generative Flows"
_NeurIPS.cc/2024/Conference — NeurIPS 2024 poster_

### Official Review · Reviewer_6n8r · 2024-07-11

**Soundness:** 3
**Presentation:** 3
**Contribution:** 2
**Rating:** 6
**Confidence:** 4

**Summary:**

This paper proposes a new matching method based on Hamiltonian mechanics. The proposed method, algorithmically, is essentially a second-order ODE in which the acceleration (i.e., the drift of the velocity channel) is approximated by deep neural networks. The method is well grounded on the theory of classical Hamiltonian mechanics and statistical physics. As a higher-order system, the proposed method naturally absorbs prior diffusion and flow methods as special cases. Experiments were conducted on a synthetic dataset (Gaussian mixtures) and image generation.

**Strengths:**

- Presentation is overall clear and easy to follow. I like how the authors highlight important sentences for improving readability. Figs 1 & 2 are also instructive in showcasing the advantage of their method.

- Theoretical contributions are solid. I think many of their results in Sec 4 may as well be of interested for readers from other domains.

**Weaknesses:**

- It's unfortunate that, from an algorithmic standpoint, the resulting algorithm---after all these nice theoretical results---remains somewhat similar to a few recently-proposed second-order methods, e.g., AGM (https://arxiv.org/pdf/2310.07805 ICLR'24 oral) or CLD (ref. [12]).

- The use of harmonic oscillator in Sec 7 is not very motivated in my opinion. There could be other alternatives that also fulfill C1 & C2 conditions. Practically speaking, sampling (x,v) in (23) from sin/cos doesn't seem to encourage straighter generative process.

- (minor) Typo in L236, I think you meant C2 not C1.

Overall, I have a mixed feeling on this paper --- on one side, I do enjoy, and acknowledge, their theoretical contributions, yet on another side, I feel like the resulting algorithm is rather weak.

**Questions:**

- Can the authors comment on the differences compared to AGM? I think AGM also constructs simulation-free (xt, vt) samples then matches the drift of the velocity.

- Have the author ever tried to jointly train both $\theta$ and $\phi$, as mentioned in L179? I understand this may most likely be impractical but just curious from an exploratory standpoint.

- Could the authors elaborate more on L175 the "optimal" choice of T, and the first hypothesis in L184?

**Limitations:**

Limitations were mentioned in Sec 10.

---

> ### Author Rebuttal · Authors · 2024-08-06
>
> We thank the reviewer for their insightful comments and for taking the time to review our work. We address your questions and comments below.
>
> > Can the authors comment on the differences compared to AGM?
>
> **Comparison with Acceleration Generative Model (AGM) model**: Thank you for pointing out the AGM paper, which was published at ICLR 2024, 1-2 months before our own submission. We will make sure to discuss it in the camera-ready version. As you point out, both AGM and our own work - as well as the mentioned critically-damped Langevin diffusion work - use constructions in phase space (joint position and velocity space). Further, they use 2nd order ODEs for their constructions, as we do. Finally, all take inspiration from concepts used in physics. However, there are important differences in the respective frameworks:
> - **Force field vs optimal velocity predictor:** AGM focuses on learning the force field (equation (10) in AGM paper). In contrast, our approach primarily focuses on learning the optimal velocity predictor. While we also consider optimizing the force field by minimizing the norm of the optimal velocity predictor, this happens in the “outer loop” of the maximization - the inner loop optimizes the optimal velocity predictor.
> - **Scope and task:** ATM focuses on bridging two desired distributions. They pose a stochastic bridge problem in phase space (equation (5) in AGM paper) that, in short, searches for the force field that optimally bridges two distributions - where optimally is framed as bridging with minimal acceleration, kinetic energy, and potential energy (weighted respectively). We do not consider the problem of bridging distributions. To illustrate the differences, we included a description of Reflection HGFs below that illustrate a model that shows the difference in the design to AGMs further.
> - **Performance:** These methodological differences are also reflected on the experimental benchmarks as our model significantly outperforms the AGM model. While AGM achieves an FID of 2.46 on unconditional CIFAR-10 generation, we achieve an FID of 2.12 with 40% less NFEs (35 NFEs vs 50 NFEs) (see table 3 of their paper).
> - **Connection to Hamiltonian physics and score metric:** The AGM framework considers dynamics changes and learns the forces that obtain desired dynamic changes. In contrast, our framework centers around energy preservation and divergence from that preservation (for optimal velocity predictors that are not zero). Specifically, we establish a connection to Hamiltonian physics and a property of the preservation of energy. This allows us to introduce a further bi-level optimization and the possibility of joint training for score matching. “Imperfect” velocity predictors allow us to build Hamiltonian Generative Flows (HGFs). We do not find such contributions to be reflected in the AGM work.
> You also mentioned the critically-damped Langevin diffusion (CLD) work that has also been discussed in our submission. The CLD paper considers a single force field and focuses on the stochastic part (choses “optimal” amount of stochasticity to be critically damped). This is also qualitatively different from our own work. Therefore, we would make the claim that AGM and CLD are different frameworks from HGFs, while showing certain similarities.
>
> > Have the author ever tried to jointly train both 𝜃 and 𝜙, as mentioned in L179?
>
> **Joint training of 𝜃 and 𝜙:** To address your comment, we have run more experiments on this and trained an energy-based model via join training of the force field (“𝜃”) and the velocity predictor (“𝜙”). As we believe your comment was of general interest, we addressed your question on this topic in our general response and with figures in the attached PDF. We welcome additional discussions.
>
> > Could the authors elaborate more on [...] the first hypothesis in L184?
>
> **Hypothesis in L184:** In traditional score matching, the score network only ever sees points from the dataset (i.e. these are the only possible inputs). This leads to unstable training (see e.g. Kingma and LeCun (2014)). In contrast, Hamiltonian Score Matching (HSM) simulates Hamiltonian trajectories with the current estimate of the score network traversing space. Effectively, the simulation of Hamiltonian trajectories augments our data. This is similar to denoising score matching (DSM), which "augments" the data by adding noise. However, DSM does not learn the score of the original data distribution but a noisy version therefore and exhibits high variance. HSM does learn the original distribution.
>
> > Could the authors elaborate more on [...] the "optimal" choice of T?
>
> **Optimal" choice of T**: We happily elaborate on this comment. The Hamiltonian Score Discrepancy depends on the choice of a time constant T>0. The role of T is to measure the discrepancy of energy preservation in the time interval [0,T). By theorem 1, T can be chosen arbitrarily as long as T>0. Naturally, the larger T, the more discretization error we accumulate by integrating the ODE. Therefore, without proposition 2, one might suggest that to measure the Hamiltonian score discrepancy one should choose T to be very small. However, proposition 2 establishes that there is a tradeoff between ODE integration error and the “signal-to-noise ratio”: it shows that the value we estimate is approximated by a parabola. This value goes to zero fast for t->0. Therefore, choosing T>0 to be very small will lead to very small values that we aim to estimate. Our signal-to-noise ratio will be detrimental. Hence, proposition 1 indicates that there is an optimal trade-off in choosing T to be high enough to give enough signal but low enough to lead to negligible ODE discretization error.
>
> Do you believe our answers and additional experiments have addressed your concerns? We welcome any additional discussions. Thank you again for taking the time to review our work.

---

> > ### Comment · Reviewer_6n8r · 2024-08-11
> >
> > I thank the authors for the reply. I've decided to keep my score.

---

### Official Review · Reviewer_XPUP · 2024-07-13

**Soundness:** 3
**Presentation:** 3
**Contribution:** 3
**Rating:** 6
**Confidence:** 3

**Summary:**

The paper proposes the Hamiltonian Score Matching framework, which is a new general framework of generative models. Inspired by the Hamiltonian dynamics in classical and statistical mechanics, the framework uses the Hamiltonian dynamics to generate data, which is also called Hamiltonian generative flow in the paper. The paper theoretically proves that the force field coincides with the score function if and only if its optimal velocity predictor is zero and proposes the HSM loss function. The paper also shows that the Flow Matching framework and the diffusion model are special cases of the HSM framework. Additionally, a new generative flow called Oscillation Hamiltonian generative flow is constructed. Experiments on several datasets show the experimental performance of the HSM framework.

**Strengths:**

1. Very clear writing.
2. The method is novel and easy to understand.
3. The paper clarifies the connection and difference with previous works.

**Weaknesses:**

1. The experimental result is not so good. As the paper writes, the Oscillations HGFs can surpass most baselines but still lack behind the EDM model. I think more experimental results should be provided to show that the HSM framework is a good framework for generative models.

**Questions:**

What is the motivation for designing Oscillation HGFs? I'm curious about how to design a force field in the HSM framework.

---

> ### Author Rebuttal · Authors · 2024-08-06
>
> We thank the reviewer for their insightful comments and for taking the time to review our work.
>
> > What is the motivation for designing Oscillation HGFs?
>
> **Motivation for designing Oscillation HGFs and other force fields:** As you point out, the choice of a force field is an important design parameter of Oscillation HGFs. We addressed the reason to choose harmonic oscillators in the general response in detail. As a brief summary: if you assume that your data distribution is normal (as a first order approximation) and if you define the Hamiltonian as in our paper (equation (1)), then the corresponding Hamiltonian force field is F(x)=-x and the Hamiltonian dynamics are harmonic oscillators. Hence, conservation of energy and an a priori approximation of our data as a normal distribution naturally leads us to using harmonic oscillators. However, one can also consider force fields of other shapes. To illustrate this, a simple force field is the “reflection force field”, i.e. the force field that just reflects a particle to stay within an interval [a,b] (usually the domain of the data). This leads to Reflection HGFs, which are explained in more detail in the general response. To illustrate this, we trained these models and illustrated them in figure 3 and 4 in the attached PDF. We will include these illustrations and derivations for Reflection HGFs as additional illustrations for the power of HGFs in the supplementary material of the camera-ready version.
>
> > I'm curious about how to design a force field in the HSM framework.
>
> **Designing a force field in the HSM framework:** Generally speaking, we suggest the following decision tree to design a force field:
> - Trainable force field: If one decides to train the force field jointly with the velocity predictor (accepting increased computational cost), then one can parameterize the force field as a neural network and train it with the proposed procedures (we have also included experiments on this in our general rebuttal and in figure 2 in the attached PDF).
> - Fixed/designed force field: Otherwise, one should explicitly design a force field.
> 	- A. Application-specific force field: If one focuses on applications in biology/chemistry/physics, one should consider a force field relevant for this application. It is important to note that data in these fields (such as in protein folding) often lies on manifolds and future work is required to adapt HGFs to geometric data.
> 	- B. Data distribution: If an application has no a priori force field, one should aim for a force field that approximately preserves the energy of the system. If we a priori approximate our data distribution with a normal distribution (only for the purposes of designing the force field), the corresponding Hamiltonian dynamics become harmonic oscillators. This motivated us to use Oscillation HGFs (this is explained in more detail in the general response). However, other data types might have other distributions that approximate them and the respective Hamiltonian dynamics should be considered then - such as Reflection HGFs.
>
> **Lacking behind the EDM model and experimental results:** We hope to have addressed your question on this topic in our general response in detail. As a brief summary, we believe that EDM has better performance because the compute budget used to optimize EDM was extensive, reducing the FID of diffusion models on CIFAR10 from >3.0 (higher than our score) to 1.97 (slightly below our score). In fact, such an extensive optimization of the design space of diffusion models was the goal of the EDM paper, while our goal was to introduce a new framework and method. We anticipate that with a similar optimization in the future, Oscillation HGFs will show a similar performance boost (see also fig 1 in attached PDF for a visual comparison of both models). We welcome additional discussions, if needed.
>
> Do you believe our answers and additional experiments have addressed your concerns? We welcome any additional discussions. Thank you again for taking the time to review our work.

---

> > ### Comment · Reviewer_XPUP · 2024-08-11
> >
> > Thanks for your reply. All of my concerns have been addressed. I'll keep my positive rating of 6.

---

### Official Review · Reviewer_8ET8 · 2024-07-13

**Soundness:** 3
**Presentation:** 3
**Contribution:** 3
**Rating:** 7
**Confidence:** 3

**Summary:**

The authors introduce Hamiltonian velocity predictors (HVPs) as a tool for score matching and generative models. They present two innovations constructed with HVPs: Hamiltonian Score Matching (HSM), a novel generative model that encompasses diffusion models, and flow matching as HGFs with zero force fields. They showcase the extended design space of force fields by introducing Oscillation HGFs, a generative model inspired by harmonic oscillators. They show experiments validating the theoretical insights about HSM.

**Strengths:**

The paper presents an interesting generative model by leveraging  Hamiltonian velocity predictors. The authors give a theoretical analysis of the proposed method and its connections with existing methods. The authors also provide an experimental study to support the proposed method.

**Weaknesses:**

a. Why does it produce inferior performance on image generation compared to EDM?  The author could present the reasons for this.

b. The authors also need to present the computation cost/efficiency comparison of different methods to validate the proposed method.

**Questions:**

See weakness.

**Limitations:**

The authors did not provide a clear discussion of the limitations of the proposed method. This should be included in the paper.

---

> ### Author Rebuttal · Authors · 2024-08-06
>
> We thank you for your insightful comments and for taking the time to review our work. We are pleased to read that you consider HGFs an “interesting generative model” and address your questions and comments below.
>
> **Explanation for EDM vs Oscillation HGFs performance:** We addressed your question on this topic in our general response. As a brief summary, we believe that EDM has better performance because it could built on previous works optimizing diffusion models and the compute budget used to optimize EDM was orders of magnitude larger, reducing the FID of diffusion models on CIFAR10 from >3.0 (higher than our score) to 1.97 (slightly below our score). In fact, such an extensive optimization of the design space of diffusion models was the goal of the EDM paper, while our goal was to introduce a new framework and method. We anticipate that with a similar optimization in the future, Oscillation HGFs will show a similar performance boost (see also fig 1 in attached PDF for a visual comparison of both models). For further explanation, we refer to the general response.
>
> **Computational details (runtime / memory):** The computational cost of Hamiltonian Generative Flows is mainly influenced by two components: (1) The cost for simulating the trajectory defined by a force field and (2) evaluating the velocity predictor. The input and output shape of the velocity predictor are the same as the denoiser or score network for diffusion. Therefore, they share the same computational cost as diffusion models and scale with the size of the network. The cost for simulating the force field depends on the choice of the force field: (A) For a trained force field (see section on Hamiltonian Score Matching), one has to backpropagate through the trajectory. In our experiments, we usually needed to simulate for ~5 time steps (so this is associated with a 5x increase in cost). (B) For a fixed force field, the cost is the one for simulating the ODE. In physical applications with known force fields, one could also use ODE solvers and the cost is application-specific. (C) In many cases (Oscillation HGFs, Reflection HGFs, diffusion models, flow matching), one can compute these in a simulation-free manner, so there is negligible additional cost and the only cost consists of forward pass through the velocity predictor.
>
> **Limitations of HGFs:** We are happy to elaborate more on the limitations of our method. We will include a discussion of the above limitations in the camera-ready version. The limitations of the method proposed in our framework are - depending on the use case - as follows:
> - **Hamiltonian Score Discrepancy:** Here, limitations are the discretization error of the ODE and the simulation cost (to compute the discrepancy score, no backpropagation through the force network is needed).
> - **Hamiltonian Score Matching via joint training:** Training through a simulated ODE requires optimizing a Neural ODE. In our experiments, this was not a big problem but generally speaking led to around 5x increase of computation cost (i.e. around 5 forward passes through the network per training evaluation).
> - **Hamiltonian Generative Flows as a generative model:** To train HGFs, the final distribution might be not known (see condition (C2)). Therefore, while the learnt ODE is always “correct”, the initial distribution, which we start our sampling process with, might not approximate the actual distribution. This error might propagate forward to samples from an unrealistic distribution. However, in many cases, one knows the distribution after a certain time up to a negligible error. To illustrate this, we developed and trained Reflection HGFs. The idea of the model is that particles can move freely in a box without collision with walls (“very strong forces”) at the boundaries of the data domain making the particles bounce back (this can be made rigorous with von Neumann boundary conditions). With normally distributed velocities, the distribution of particles will converge towards a uniform distribution. Therefore, even if we cannot compute the final distribution analytically, we often might still know it approximately.
>
> Together with our general response, we hope that the above addresses all of your comments. We welcome any additional discussions.

---

> > ### Comment · Reviewer_8ET8 · 2024-08-12
> >
> > Thank the authors for the response. I will keep the score unchanged.

---

### Official Review · Reviewer_WwWf · 2024-07-28

**Soundness:** 3
**Presentation:** 3
**Contribution:** 2
**Rating:** 6
**Confidence:** 3

**Summary:**

In this work, the authors proposed a new generative modeling approach called Hamiltonian Score Matching, which is motivated by classical Hamiltonian mechanics. This approach estimates score functions by augmenting data via Hamiltonian trajectories, and further motivates Hamiltonian Generative Flows. The authors also discuss the design space of force fields in Hamiltonian Generative Flows and connect it with harmonic oscillators. Experiments are conducted to verify the effectiveness of the proposed approach.

**Strengths:**

1. The connection between Hamiltonian dynamics, force fields, and score matching is interesting and provides a new perspective for the design space of generative models.

2. The paper is well-written and easy to follow.

3. The authors provide a clear proof section to check the correctness of propositions and theorems.

**Weaknesses:**

1. Regarding the methodology: The authors spend efforts to establish Hamiltonian Generative Flows from Parametrized Hamiltonian ODEs, Hamiltonian velocity predictor, and Theorem 1 for Hamiltonian Score Matching.   Although the benefits such as bringing additional freedoms for force fields and prediction objectives, the scalability of this approach is restricted as stated by the authors. The necessary trajectory simulation and min-max optimization largely limit the usage of the proposed approach. In Section 7, the authors instead seek to use a pre-defined force field instead of parametrized models, which makes the HGFs degenerate to pre-defined ODEs, further indicating the theory-practice gap of the proposed approach.

2. Regarding the experiments: scales and types of experiments in this work are rather limited. In addition to the simulation experiment, the authors only conducted experiments on CIFAR-10, which even shows that the proposed HGFs cannot bring explicit benefits compared to previous approaches. It would largely improve the quality of this work if the authors could provide more diverse and strong experimental results for the proposed approach.

Overall, I think the paper provides an interesting perspective, but the current issues discount the ratings of my evaluation. Currently, I vote for borderline acceptance and I will carefully read the rebuttal and other reviews to decide whether to increase or decrease my scores.

**Questions:**

See the above section.

**Limitations:**

The authors discussed the limitations of this work in the Conclusion section.

---

> ### Author Rebuttal · Authors · 2024-08-06
>
> We thank you for your insightful comments and for taking the time to review our work. We decided to address your questions regarding our methodology and our experimental results in the general response. We provide additional information here.
>
> **Theory vs practice:** You correctly point out that there is a trade-off between “theory and practice”. We consider elucidating this trade-off as one of the main contributions of our work. We showcase that score matching corresponds to “perfect” energy conservation but leads to a “harder” training objectives, while violating energy conservation allows us to make the algorithms scalable under the conditions we outlined. We introduced the Hamiltonian Score Matching (HSM) method first because by theorem 1, it is the “ideal” limit case one would like to learn. Despite a min-max objective, HSM provides a novel way of learning the score of a data distribution without adding noise (denoising score matching) or requiring to estimate the trace of the Hessian (implicit score matching). In Fig (2) in the attached PDF, we illustrate that energy-based models can be trained with this objective and lead to high-quality samples. However, we agree with your assessment that the proposed min-max optimization required for HSM might be an issue for training stability. For this reason, we then introduced Hamiltonian Generative Flows (HGFs) to be a scalable alternative but one that requires us to take into account that our dynamics do not necessarily conserve the energy. This, as you point out, leads us to use pre-defined force fields. The design space of pre-defined force fields is already huge and choosing the right force field is equally informed by the ideal “limit” of Hamiltonian Score Matching. For example, Oscillation HGFs correspond to Hamiltonian dynamics under a normal approximation of the data (see general response for details). There are many other interesting examples such as Reflection HGFs (see paragraph below).
>
> **Extended design space - Reflection HGFs:** To further illustrate the power of the HGF framework and that pre-defined ODEs can lead to novel interesting models, we developed “Reflection HGFs” here and plot the result of training them in figure 3 and figure 4 in the attached PDF. Let us assume that we a priori approximate our data as a uniform distribution over its support. Aiming for energy conservation, we consider the force field associated with a uniform distribution in [0,1]. This corresponds to zero force in (0,1) but a reflection (i.e. flip of the sign of the velocity) on the boundaries of the domain. A reflection can be considered an infinite force at the boundary (one can also use Neumann boundary conditions to make this precise and not an asymptotic limit). If we simulate a particle with a normally distributed velocity and let it bounce around for a long time, its location will eventually become approximately uniformly distributed. Therefore, starting with an initial distribution “p_data x N(0,I)”, we end up with a distribution of locations at time T that is uniform if T is large enough. Therefore, condition (C2) in the definition of HGFs is satisfied and condition (C1) is also satisfied as we can compute loss in a simulation-free way. Learning the Hamiltonian Velocity Predictors and running the backwards ODE allows us to generate data. We trained such a model on a simple toy distribution and illustrated it in figure 3 and figure 4 in the attached PDF. Such a model is inspired from the ideas of energy conservation developed in our work, and we hope, illustrates that HGFs are different from various previous models and not restricted to only Oscillation HGFs, diffusion models, or flow matching.
>
> **Performance comparison to EDM:** We addressed your question on this topic in our general response.
> As a brief summary, we believe that EDM has better performance because it could build on previous works optimizing diffusion models and the compute budget used to optimize EDM was orders of magnitude larger, reducing the FID of diffusion models on CIFAR10 from >3.0 (higher than our score) to 1.97 (slightly below our score). In fact, such an extensive optimization of the design space of diffusion models was the goal of the EDM paper, while our goal was to introduce a new framework and method. We anticipate that with a similar optimization in the future, Oscillation HGFs will show a similar performance boost (see also fig 1 in attached PDF for a visual comparison of both models). For further explanation, we refer to the general response.
>
> **Experiments and choice of datasets:** We would like to draw attention to the fact that we not only conducted experiments on CIFAR-10 32x32, as suggested, but also on the FFHQ 64x64 dataset (see figure 4 in our paper). We chose CIFAR10 because of its use as a standard benchmark on image generation. To test our model on a higher resolution and more realistic dataset, we chose the FFHQ 64x64 dataset. The reason for this choice was that it provides higher resolution images at a dataset size that is possible to still benchmark on a compute budget available in academia. For example, ImageNet training would take 32 days on 8 A100 GPUs (this is the training time for the EDM model that was used). We also agree with your assessment that experiments on a physical dataset might have been advantageous. In particular, we considered molecular machine learning benchmarks such as ligand docking or protein design. Unfortunately, these modalities often require data to lie on manifolds (usually SE(3)) and require models to be designed with equivariance as an implicit bias. Designing generative models for this task are usually separate works as they require extensive design adaptations and ablations. We considered this as out of scope for this work.
>
> Together with our general response, we hope that the above addresses all of your comments. We welcome any additional discussions. Thank you again for your helpful feedback and response.

---

> ### Comment · Reviewer_WwWf · 2024-08-10
>
> Thank you for your clarifications. Most of my concerns have been addressed. I choose to increase my rating to 6.

---

### Official Review · Reviewer_g7vb · 2024-07-29

**Soundness:** 4
**Presentation:** 3
**Contribution:** 4
**Rating:** 8
**Confidence:** 3

**Summary:**

The paper explores the application of Hamiltonian formalism for generative modelling.
In this framework, the score function is interpreted as a force field F, thus optimizing the parameters of F yields a score-matching objective.
Authors build the relation to Flow Matching and Diffusion models as special cases of their framework and suggest another special case, which they validate on an image generation task.

**Strengths:**

- the formulation is novel and implies conservation of energy and volume.
- the theoretical foundation and contribution of the work is very solid.
- the authors derive diffusion models and flow matching as a special case of HGFs.
- the experimental results are solid and indicate the impressive capabilities of the framework.

**Weaknesses:**

- in the general case, the framework requires joint learning of two functions: force field and velocity.
The authors themselves compare the optimization to GANs, which raises the question of training stability.
- the authors do not compare the runtime and memory consumption of the framework with other SOTA methods, e.g. Flow Matching.
- the experimental section is somewhat limited to a toy experiment and a single image-generation task.
It would be highly beneficial to evaluate the framework on another task (e.g. physics-related), although it is not critical given the high theoretical contribution of the work.
- it is not clear how image generation is influenced by Hamiltonian dynamics. What is x, and what is v?

**Questions:**

- I am curious if it would be simple to enforce equivariance as in Equivariant Hamiltonian Flows by Rezende et al.
- as the force field form is a choice now, did authors consider alternative forms other than osciallators? I am curious what is the intuition behind the choice.
- authors mentioned conservation properties in the context of physical simulations.
Do you think that they will still be beneficial in the general case, e.g. image/video generation?
Even in physical systems, if the energy is not conserved, would those properties be a limiting factor somehow?

**Limitations:**

Perhaps the main limitation for me is the discussion of the framework's theoretical properties, e.g., conservation of energy and volume, in the context of generative modelling. It would be beneficial to see an analysis of how those inductive biases will play out in specific cases (e.g., physical simulations or image generation). Besides, it would be beneficial to provide computational details (runtime / memory) and compare them to SOTA methods.

---

> ### Author Rebuttal · Authors · 2024-08-06
>
> We thank the reviewer for insightful comments and for taking the time to review our work.
>
> > I am curious if it would be simple to enforce equivariance as in Equivariant Hamiltonian Flows by Rezende et al.
>
> **Enforcing equivariance:** Thank you for highlighting the work “Equivariant Hamiltonian Flows”. In this work, the authors modify an ELBO training objective as a constrained optimization such that the learnt distribution preserves symmetries. While such an idea could in principle be also applied to HGFs, it is not as straight-forward due to differences between “traditional” continuous normalizing flows (CNFs) and our approach:
> (1) Equivariant Hamiltonian Flows employ maximum likelihood estimation (via an ELBO, equation (1) in their paper) to train CNFs, while we consider velocity prediction as an objective.
> (2) CNFs learn to transform noise to data in an arbitrary fashion, while HGFs fix the probability path (if the force field is fixed)
> One would need to account for these differences by designing the force field in an equivariant manner. Depending on the type of equivariance considered, one might want to consider techniques as in “Equivariant flow matching” that can similarly be extended from flow matching to HGFs by introducing equivariances in the design of the force field.
>
> > As the force field form is a choice now, did authors consider alternative forms other than osciallators? I am curious what is the intuition behind the choice.
>
> **Choices of force fields other than harmonic oscillators:** As you point out, the choice of a force field is an important design parameter of HGFs. We, indeed, can consider other force fields different from Oscillation HGFs. For example, a simple force field is the “reflection force field”, i.e. the force field that just reflects a particle to stay within an interval [a,b] (usually the domain of the data). This leads to Reflection HGFs. To address your comment, we implemented these and illustrated them in figure 3 and 4 in the attached PDF. We would be happy to include these illustrations and derivations for Reflection HGFs as additional illustrations in the supplementary material of the camera-ready version. Finally, we chose Oscillation HGFs for our benchmarks because they are based on harmonic oscillators and dynamics come with beneficial properties. For example, harmonic oscillators are natural dynamics to consider under a normal approximation of our data as they correspond to Hamiltonian dynamics under this approximation. Further, they appeal due to their simplicity. We also addressed the motivation for Oscillation HGFs in more detail in our general response.
>
> > Authors mentioned conservation properties in the context of physical simulations. Do you think that they will still be beneficial in the general case, e.g. image/video generation? Even in physical systems, if the energy is not conserved, would those properties be a limiting factor somehow?
>
> **Physical simulations and conservation properties:** In physical systems, enforcing conservation properties serves as an inductive bias, similar to equivariance, and such biases have been observed to improve training accuracy. We anticipate that the same would be true of energy conservation. Conservation properties in the context of non-physical applications would also be beneficial in our opinion. As pointed out in the previous paragraph, Oscillation HGFs correspond to Hamiltonian dynamics under a normal approximation of the data. The Hamiltonian in this case is given by a sum of the squared norm of the velocity and the squared norm of the data point. This corresponds to the squared norm of the joint vector (x,v) in phase space and preserving this means that the dynamics always have constant scale. This ensures training stability. In fact, for our training we did not employ any rescaling of inputs/outputs, skip connections, or other preconditioning done to make training more stable for diffusion models (see e.g. the EDM paper). Oscillation HGFs achieved high performance with very little fine-tuning. We consider this as evidence that the conservation properties of the dynamics lead to better training dynamics.
>
> > What is x, and what is v?
>
> **What is x, and what is v?** Next, we address your question about what “x” and “v” stand for in the image generation case. For image generation, x is the image (i.e. a data point) and v is an auxiliary variable (initialized with a normal distribution). We will make sure to highlight this in a camera-ready version.
>
> **Computational details (runtime / memory):** The computational cost of Hamiltonian Generative Flows is mainly influenced by two components: (1) The cost for simulating the trajectory defined by a force field and (2) the cost of evaluating the velocity predictor. The input and output shape of the velocity predictor are the same as the denoiser or score network for diffusion. Therefore, they share the same computational cost as diffusion models and scale with the size of the network. The cost for simulating the force field depends on the choice of the force field: (A) For a trained force field (see section on Hamiltonian Score Matching), one has to backpropagate through the trajectory. This can be done with constant in memory using common techniques for Neural ODEs. In our experiments, we usually needed to simulate for ~5 time steps (so this is associated with 5 evaluations of the force field). (B) For a fixed force field, the cost is the one for simulating the ODE. In physical applications with known force fields, one could also use ODE solvers and the cost is application-specific. (C) In many cases (Oscillation HGFs, Reflection HGFs, diffusion models, flow matching), one can compute these in a simulation-free manner, so there is negligible additional cost.
>
> Together with our general response, we hope that the above addresses all of your comments. We welcome any additional discussions. Thank you again for your helpful feedback and response.

---

> > ### Comment · Reviewer_g7vb · 2024-08-10
> > **Official Comment by Reviewer g7vb**
> >
> > I thank the authors for their response and additional experiments, which certainly prove the positive contribution of the submission.
> > I will retain my score.

---

### Author Rebuttal · Authors · 2024-08-06

We would like to thank all reviewers for their constructive and positive feedback. We are pleased to see that our work is considered by reviewers as a “very solid theoretical contribution” introducing an “interesting generative model leveraging Hamiltonian velocity predictors”. Below, we’ve compiled rebuttal points asked by several reviewers while we address individual questions in the individual rebuttals. We welcome any additional discussions.

**Motivation for Oscillation HGFs**: One of the main instantiations of Hamiltonian Generative Flows (HGFs) that we study empirically are Oscillation HGFs (see section 7), an instantiation of HGFs relying on harmonic oscillators. Multiple reviewers asked us to elaborate more on the motivation to study Oscillation HGFs. We studied them for several reasons:
- **Oscillation HGFs are Hamiltonian dynamics under a normal approximation of data:** If you assume that the data distribution is Gaussian N(0,1) (as a first order approximation) and if you define the Hamiltonian as in our paper (equation (1)), then the corresponding Hamiltonian force field is F(x)=-x and the Hamiltonian dynamics are harmonic oscillators. Hence, conservation of energy and an a priori approximation of our data as a normal distribution naturally leads us to using harmonic oscillators. (The normal approximation is only used for the purposes of defining the force field and does not restrict the actual generated distribution to be normal).
- **Simplicity and analytical tractability:** The harmonic oscillator is one of the simplest linear systems with closed-form solutions, necessary to train the model “simulation-free”. The only simpler force field would be the zero force field, which, as shown, corresponds to diffusion models.
- **Training stability:** All n-order derivatives of the harmonic oscillator have constant expected norm (eq. (25)). We hypothesize that this leads to more stable training. In fact, for our training we did not employ any rescaling of inputs/outputs, skip connections, or other preconditioning. Oscillation HGFs achieved high performance with very little fine-tuning.

**Comparison to EDM paper:** Multiple reviewers asked for a comparison with the EDM model and for the reasons why EDM has slightly better performance at this point in time:
- **Scope of work:** We consider our work as introducing a novel meta-algorithm and the experiments serve as an illustration of its power. The EDM network is a diffusion model and therefore, a specific instance of HGFs. We believe that our framework offers potential to find new well-performing methods within the framework. For example, Oscillation HGFs gave a model that - almost out of the box - performs almost on par with EDM.
- **Hyperparameter tuning:** The EDM paper used an extensive compute budget to tune a variety of hyperparameters such as noise schedule, architectures, noise distributions during training, loss weightings, network preconditioning, learning hyperparameters, and many more - all optimized for CIFAR10. In fact, this was the goal of their EDM, while our paper has a methodological goal. This reduced their FID score on CIFAR-10 from >3.0 (a score higher than ours) to 1.98.  We anticipate that with similar optimization as done in EDM, Oscillation HGFs might perform even better.
- **Methodological comparison:** Another question was whether EDM paths are straighter/smoother than Oscillation HGF paths. We believe that this question was inspired by the visualization in figure 3 in the EDM paper. To address this question, we replicated this visualization for Oscillation HGFs (see figure 1 in the attached PDF). As one can see, EDM paths have stronger curvature at the beginning, while Oscillation HGFs have stronger curvature mid-way (0.4<t<0.8). Hence, it is unclear whether this provides a meaningful difference.  Please note that the results of these visualizations might not correspond to sampling performance as they do not depict the marginal “learnt” velocity predictor but only conditioned on a few data points.

**Joint training of 𝜃 and 𝜙:** Multiple reviewers asked for the possibility of training an energy-based model via joint training of the velocity predictor and the force field at the same time (we had shown how to learn a score network in figure 1 in our paper already). This corresponds to training an energy-based model via Hamiltonian score matching and sampling via Langevin dynamics or Hamiltonian Monte Carlo. To address this comment, we have trained an energy-based model on a 2d Gaussian mixture with Hamiltonian Score Matching and sampled from it. In figure 2 in the attached PDF, one can see that the generated samples closely match the desired distribution. As mentioned in the paper and as reviewer 6n8r points out, such a model is more of theoretical interest because of the required min-max optimization.

**Reflection HGFs:** Several reviewers indicated that they would be interested in whether HGFs can give us models other than Oscillation HGFs. To address this, we illustrate this possibility by introducing “Reflection HGFs” here and plot the result of training them in figure 3 and figure 4 in the attached PDF. The idea of the model is that particles can move freely in a box without collision with walls (“very strong forces”) at the boundaries of the data domain making the particles bounce back (this can be made rigorous with von Neumann boundary conditions). With normally distributed velocities, the distribution of particles will converge towards a uniform distribution. Further, this model can be trained in a simulation-free manner. We trained this HGF model on a simple toy distribution (see fig 3 and fig 4 in the attached PDF). Such a HGF model is distinct from previous models and, we hope, illustrates that HGFs are not restricted to Oscillation HGFs, diffusion models, or flow matching.

We thank the reviewers again for their helpful comments. We welcome any additional discussions.

---

### Comment · Area_Chair_cXA1 · 2024-08-08
**Author-Reviewer Discussion Phase (ends Aug 13 midnight AOE)**

Dear reviewers,

Thank you for your efforts so far! The authors have responded to each of your reviews and have also provided some additional results in the PDF attached to the overall response. Could those you take a moment to read the author response and follow up regarding any points that may further inform your opinion on the paper?

With thanks, AC

---

### Decision · Program_Chairs · 2024-09-25

**Decision:**

Accept (poster)

**Comment:**

This work develops methods for generative modeling based on Hamiltonian dynamics. In this setup the authors combine a learned force field (which approximate the score function of the data distribution) with an auxiliary velocity variable, for which they define a learned Hamiltonian velocity predictor (HVP). The HVP can be used to define a Hamiltonian score matching method, which then leads to the definition of Hamiltonian Generative Flows (HGFs), a class of generative models that recover diffusion models and flow matching as special cases (with zero force field).

Reviewers are in agreement that this is a clearly paper presents an interesting new Hamiltonian perspective on generative modeling. Reviewer do note some weaknesses in experiments, notably the slightly poor performance relative to EDM, though reviewers do not appear to find this a key consideration. Overall, this is a submission that is above the bar for acceptance. In the camera ready, do please make sure to discuss the connections with the AGM method as pointed out by reviewer 6n8r.